# Offline Reinforcement Learning Through Trajectory Clustering and Lower Bound Penalisation

## Abstract

In this paper, we propose a new framework for value regularisation in offline reinforcement learning (RL). While most previous methods evade explicit out-of-distribution (OOD) region identification due to its difficulty, our method explicitly identifies the OOD region, which can be non-convex depending on datasets, via a newly proposed trajectory clustering-based behaviour cloning algorithm. With the obtained explicit OOD region, we then define a Bellman-type operator pushing the value in the OOD region to a tight lower bound while operating normally in the in-distribution region. The value function with this operator can be used for policy acquisition in various ways. Empirical results on multiple offline RL benchmarks show that our method yields the state-of-the-art performance.

## 1 Introduction

Offline reinforcement learning (RL) has attracted significant attention due to its sample efficiency and safety. Unlike conventional RL, where an agent learns an optimal policy through interactions with the environment, offline RL disallows any environmental interactions. Instead, the agent is provided with a fixed dataset $\mathcal{D}$ of trajectories and should derive a competent policy solely from these samples.

Although offline RL also relies on off-policy data, standard off-policy RL algorithms often fail in this setting. The primary cause is the extrapolation error of the critic, which cannot be corrected, as the agent is unable to re-evaluate overestimated out-of-distribution (OOD) actions through environmental interactions (Fujimoto et al., 2019). These errors not only persist but also accumulate through bootstrapping, making careful handling of OOD actions crucial for stable training.

Value regularisation has emerged as one of the main strategies for addressing the extrapolation issue (Kumar et al., 2020; Lyu et al., 2022; Mao et al., 2023). By penalising the critic values of OOD actions, these methods encourage the agent to prefer in-distribution (ID) actions over OOD ones. A variety of algorithms have been proposed within this paradigm, differing mainly in their choice of regularisation term. However, most rely on indirect proxies to determine the OOD-ness of actions. For example, Mao et al. (2023) approximates the behaviour policy with a Gaussian model to compute importance-sampling ratios. When the dataset contains multiple disparate behaviour policies, such unimodal approximation misrepresent the underlying multimodal structure. They assign spuriously high densities to the inter-modal region, distorting the OOD-ness estimates, eventually degrading the regularisation's effectiveness.

To address these limitations, we propose a novel value-regularisation algorithm that explicitly identifies the OOD region and penalises the critic's estimates within those regions using a newly derived lower bound that is tighter than those employed in previous work. Empirical evaluations on the D4RL benchmark (Fu et al., 2020) demonstrate that our approach yields high-performing policies across a wide range of offline RL datasets.

**Our main contributions are:**

- **Likelihood-based OOD detection.**
  We introduce a principled likelihood-based criterion for identifying the OOD region.

- **Behaviour policy modelling via trajectory clustering.**
  By linking trajectory clustering in offline RL to task identification in meta RL, we propose an algorithm that learns a Gaussian mixture model of the dataset, enabling accurate density estimation in multimodal datasets.
- **A tight lower-bound-based value regulariser.**
  We theoretically derive a tight lower bound on the optimal action-value function and incorporate it into a value regulariser.

## 2 BACKGROUND

**Notation** For the list of notations used in this paper and their meanings, refer to Appendix Sec. A.

**Markov Decision Process** An RL problem is formulated as a Markov Decision Process (MDP), which is defined as a 6-tuple $\mathcal{M} = \langle \mathcal{S}, \mathcal{A}, P, R, \gamma, \rho_0 \rangle$, where $\mathcal{S} \subseteq \mathbb{R}^{d_s}$ is the state space, $\mathcal{A} \subseteq \mathbb{R}^{d_a}$ is the action space, $P \colon \mathcal{S} \times \mathcal{A} \to \mathcal{P}(\mathcal{S})$ is the transition dynamics, $R \colon \mathcal{S} \times \mathcal{A} \times \mathcal{S} \to \mathcal{P}(\mathbb{R})$ is the reward function, $\gamma \in [0, 1]$ is the discount factor, and $\rho_0 \in \mathcal{P}(\mathcal{S})$ is the initial state distribution. We will assume that the support of $R(s, a, s')$ is bounded above by $r_{\max}$ and bounded below by $r_{\min}$ for all $s, s' \in \mathcal{S}$ and $a \in \mathcal{A}$.

**Value Functions** Given a policy $\pi$, the Bellman operator $\mathcal{T}^\pi$ on $L^\infty(\mathcal{S} \times \mathcal{A})$ is defined by the following equation:

$$(\mathcal{T}^\pi Q)(s, a) = \mathbb{E}_{s' \sim P(s,a)} \left[ \mathbb{E}_{r \sim R(s,a,s')}[r] \right] + \mathbb{E}_{s' \sim P(s,a)} \left[ \mathbb{E}_{a' \sim \pi(s')} \left[ Q(s', a') \right] \right].$$

Then, the action-value function (or Q-function) $Q^\pi \colon \mathcal{S} \times \mathcal{A} \to \mathbb{R}$ is defined as the unique fixed point of $\mathcal{T}^\pi$, and the state-value function $V^\pi \colon \mathcal{S} \to \mathbb{R}$ is given by $V^\pi(s) = \mathbb{E}_{a \sim \pi(s)}[Q^\pi(s, a)]$. The objective of RL is to find an optimal policy $\pi^*$ such that $V^{\pi^*} \succeq V^\pi$ for any policy $\pi$.

**Offline Reinforcement Learning** For offline RL, interactions with the environment is prohibited, and the agent has to learn a policy from a given dataset $\mathcal{D}$ of trajectories. Throughout this paper, we will assume that each trajectory $\tau \in \mathcal{D}$ is sampled with a uni-modal behaviour policy $\beta \in \{\beta_0, \beta_1, \beta_2, \ldots, \beta_{K-1}\}$, where the candidate set $\mathcal{B} = \{\beta_0, \beta_1, \beta_2, \ldots, \beta_{K-1}\}$ is fixed but unknown to the agent.

## 3 MOTIVATION

Critic penalization or value regularisation penalises the Q-values for OOD actions, while minimizing the temporal difference error for in-distribution (ID) actions. We may formulate it with the following equation

$$\min_Q \mathbb{E}_{(s,a) \sim \mathcal{D}} \left[ (Q(s, a) - \mathcal{T}^\pi Q(s, a))^2 \right] + \mathfrak{R},$$

where $\mathfrak{R}$ is a regularizer. A crucial requirement of the regularizer is that it should be able to discriminate between ID and OOD actions since we only want to penalise the values of OOD actions. One

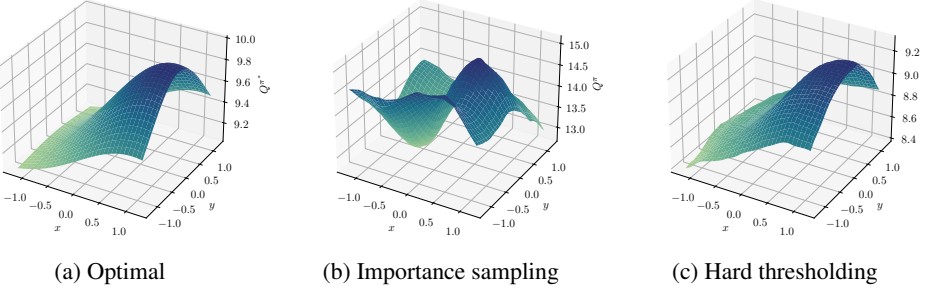

(a) Optimal      (b) Importance sampling      (c) Hard thresholding

Figure 1: The Q-values on the plane spanned by $\mathbf{e}_1$ and $\mathbf{e}_2$ estimated by each method, i.e., $Q(x\mathbf{e}_1 + y\mathbf{e}_2)$. Due to the high variance of importance sampling ratios, the importance sampling method fails to approximate the optimal Q-function accurately.

of the first approaches was to set the regulariser as (Kumar et al., 2020)

$$\mathfrak{R} = \mathbb{E}_{s \sim \mathcal{D}, a \sim \mu}[Q(s,a)] - \mathbb{E}_{s \sim \mathcal{D}, a \sim \beta}[Q(s,a)],$$

where $\beta$ is the behaviour policy and $\mu$ is some distribution that satisfies the condition $\operatorname{supp} \mu \subseteq \operatorname{supp} \beta$ (Kumar et al., 2020). While minimising the Q values for OOD actions sampled by $\mu$, they simultaneously maximised the Q values for ID actions sampled from $\beta$ to compensate for over-penalisation. However, as Mao et al. (2023) points out, this approach has two shortcomings: (i) the requirement $\operatorname{supp} \mu \subseteq \operatorname{supp} \beta$ may not hold in general; and (ii) if the dataset contains a large portion of suboptimal actions, their Q values would be overestimated. To address these issues, they proposed an importance sampling (IS)-based method that utilises the following regulariser:

$$\mathfrak{R}_{\mathrm{IS}} = \mathbb{E}_{s \sim \mathcal{D}, a \sim \mu}[(Q(s,a) - Q_{\mathrm{targ}}(s,a))^2] - \mathbb{E}_{s \sim \mathcal{D}, a \sim \beta}\left[\frac{\mu(a \mid s)}{\beta(a \mid s)}(Q(s,a) - Q_{\mathrm{targ}}(s,a))^2\right],$$

where $\mu$ is a probability distribution supported on the entire action space and $Q_{\mathrm{targ}}$ is a regulariser target, which they set to $r_{\min}/(1-\gamma)$ for all $s \in \mathcal{S}, a \in \mathcal{A}$. Since the two terms cancel each other on $\operatorname{supp} \beta$, $\mathfrak{R}_{\mathrm{IS}}$ is equivalent to $\mathbb{E}_{s \sim \mathcal{D}, a \sim \mu}\left[\mathbf{1}_{\mathcal{A} \setminus \operatorname{supp} \beta}(Q(s,a) - Q_{\mathrm{targ}}(s,a))^2\right]$, which corresponds to the goal of penalising the Q values of OOD actions.

A significant drawback of $\mathfrak{R}_{\mathrm{IS}}$ is that IS ratios are known to have high variance, especially for high-dimensional spaces. Consider a simple single-state infinite-horizon MDP with a six-dimensional action space, and an offline RL dataset of size $1\,000\,000$ sampled from a behaviour policy $\mathcal{N}(\mathbf{0}, \boldsymbol{I}_6)$. Suppose the optimal action is $\mathbf{a}^* = \mathbf{e}_1$. Then the IS ratio between $\mu = \mathcal{N}(\mathbf{a}^*, 0.04\boldsymbol{I}_6)$ and $\beta$ of the samples in the dataset ranges from $1.88 \times 10^{-225}$ to $1.93 \times 10^4$. As demonstrated in Figure 1b and Table 1, the IS method yields an inaccurate Q value estimation and a suboptimal policy due to this severe fluctuation of IS ratios.

Table 1: The discounted return of the policies learned with each method. IS stands for importance sampling and HT stands for hard thresholding.

| Optimal | IS | HT |
|---------|-----|-----|
| 10 | $2.44 \pm 0.83$ | $9.72 \pm 0.14$ |

To overcome these limitations of the previous value regularization methods, we here propose to explicitly identify the set of OOD actions $\mathrm{OOD}(s)$ for each state $s \in \mathcal{S}$, and set the regulariser to zero for ID actions. Such hard thresholding (HT) allows a more stable training process, resulting in a more accurate Q value estimations and better-performing policies, as seen in Fig. 1c and Table 1.

## 4 Proposed Method

This section is structured as follows. In Section 4.1, we first discuss how we can compute the set $\mathrm{OOD}(s)$. We then propose a new lower bound of $Q^{\pi^*}$ and show its effectiveness as a penalisation target in Section 4.2. Finally, we provide a practical offline RL algorithm in Section 4.3.

### 4.1 Identifying the Out-of-Distribution Action Set

Likelihood is the most natural way to measure how OOD a particular sample is. However, choosing the threshold value is not trivial. For blunt distributions, we should use a lower threshold value, whereas for sharp distributions, we can choose a higher value. We propose a systematic method of setting the threshold value by adopting the concept of *highest density region* (HDR; Hyndman 1996), which is basically a generalisation of a confidence interval to multivariate random variables.[1]

**Definition 1** (Hyndman 1996). *Let $f(X)$ be the pdf of a random variable $X$. Then, the $100(1-\alpha)\%$ highest density region (HDR) is the subset $\mathcal{R}(f_\alpha)$ of the sample space of $X$ such that $\mathcal{R}(f_\alpha) = \{\, x : f(x) \geq f_\alpha \,\}$, where $f_\alpha = \sup\{\, y : \mathbb{P}(X \in \mathcal{R}(y)) \geq 1 - \alpha \,\}$.*

In the following subsections, we discuss how to compute the HDR under different assumptions.

---

[1]We provide a diagram (Figure 10) showing the $100(1-\alpha)\%$ HDR of a normal distribution on page 28 to aid the understanding of the concept of a HDR.

### 4.1.1 Homogeneous Datasets

We first discuss the case when the offline dataset $\mathcal{D}$ is homogeneous, that is, it was generated from a single uni-modal behaviour policy $\beta$. Then, we may obtain a fairly accurate Gaussian approximation $\hat{\beta}$ of $\beta$ through behaviour cloning. Let $\boldsymbol{\mu}\colon \mathcal{S} \to \mathbb{R}^{d_a}$ and $\boldsymbol{\Sigma}\colon \mathcal{S} \to \mathbb{R}^{d_a \times d_a}$ be the mean and covariance matrix functions of $\hat{\beta}$, respectively. Assuming $\boldsymbol{\Sigma}(s)$ is positive definite for all $s \in \mathcal{S}$, the $100(1-\alpha)\,\%$ HDR has the following closed-form representation (Proposition 4 in Appendix):

$$\mathcal{R}_{\hat{\beta}}(f_\alpha; s) = \left\{ \mathbf{x} \in \mathbb{R}^{d_a} : A_{\hat{\beta}}(\mathbf{x}; s) \leq F_{\chi_{d_a}^2}^{-1}(1-\alpha) \right\},$$

where $F_{\chi_{d_a}^2}^{-1}$ is the inverse cumulative distribution function of a chi-squared random variable with $d_a$ degrees of freedom and

$$A_{\hat{\beta}}(\mathbf{x}; s) = (\mathbf{x} - \boldsymbol{\mu}(s))^\top \boldsymbol{\Sigma}(s)^{-1}(\mathbf{x} - \boldsymbol{\mu}(s)).$$

Choosing an appropriate value of $0 < \alpha < 1$, we can define $\mathrm{OOD}(s)$ as

$$\mathrm{OOD}(s) = \mathcal{A} \setminus \mathcal{R}_{\hat{\beta}}(f_\alpha; s). \tag{1}$$

### 4.1.2 Heterogeneous Datasets

The definition of $\mathrm{OOD}(s)$ for a homogeneous dataset given in (1) can be generalised to the heterogeneous case as

$$\mathrm{OOD}(s) = \mathcal{A} \setminus \left( \bigcup_{\beta \in \mathcal{B}} \mathcal{R}_{\beta}(f_\alpha; s) \right)$$

for $\mathcal{B}$, where $\mathcal{B}$ is the behaviour policy candidate set. If we could identify and isolate all of the trajectories in the dataset sampled from a particular behaviour policy $\beta \in \mathcal{B}$, then obtaining an estimation $\hat{\beta}$ of $\beta$ is straightforward by applying a behaviour cloning algorithm on those isolated trajectories. Then, with the estimated $\hat{\mathcal{B}} = \{\hat{\beta}_0, \hat{\beta}_1, \ldots, \hat{\beta}_{K-1}\}$, we could compute $\mathrm{OOD}(s)$ for each state $s$ as above. Therefore, in the rest of this section, we will propose how to cluster the trajectories. Note that the proposed clustering algorithm is useful not only for value regularization here but also for other offline real-world data analysis.

Our key idea is that the trajectory clustering problem closely resembles the task inference problem in meta RL. For each policy $\pi$, there is a corresponding Markov reward process (MRP) $\mathcal{M}^\pi = \langle \mathcal{S}, P^\pi, R^\pi, \gamma \rangle$, where for all $s, s' \in \mathcal{S}$ and $r \in \mathbb{R}$, the transition probability function $P^\pi\colon \mathcal{S} \to \mathcal{P}(\mathcal{S})$ and the reward probability function $R^\pi\colon \mathcal{S} \times \mathcal{S} \to \mathcal{P}(\mathbb{R})$ are defined by the equations

$$P^\pi(s' \mid s) = \mathbb{E}_{a \sim \pi(\cdot \mid s)}\left[P(s' \mid s, a)\right], \tag{2}$$

$$R^\pi(r \mid s, s') = \mathbb{E}_{a \sim \pi(\cdot \mid s)}\left[R(r \mid s, a, s')\right], \tag{3}$$

respectively. Since the dataset $\mathcal{D}$ can then be viewed as a collection of trajectories, where each trajectory is sampled from one of the MRPs $\mathcal{M}^{\beta_0}, \mathcal{M}^{\beta_1}, \mathcal{M}^{\beta_2}, \ldots, \mathcal{M}^{\beta_{K-1}}$, trajectory clustering task can be viewed as an MRP inference problem. As this formulation is almost equivalent to the MDP inference problem setting in meta RL, we infer the MRP instead of the MDP and apply a technique similar to variational Bayes-adaptive deep RL (variBAD; Zintgraf et al. 2021).

Our goal is to infer the behaviour policy index given a trajectory. To achieve this objective, we represent the index as a discrete latent variable $m$ supported on $[K] = \{0, 1, \cdots, K-1\}$ and write

$$P^{\beta_m}(s) \approx P(s\,; m), \qquad R^{\beta_m}(s, s') \approx R(s, s'\,; m), \qquad \beta_m(s) \approx \beta(s\,; m),$$

for all $s, s' \in \mathcal{S}$, sharing $P$, $R$, and $\beta$ across trajectories. The marginal pdf of a trajectory $\tau_{:T} = (s_0, a_0, r_0, s_1, a_1, r_1, s_2, a_2, r_2, \ldots, s_{T-1}, a_{T-1}, r_{T-1}, s_T)$ is

$$p(\tau_{:T}) = \rho_0(s_0) \sum_{m=0}^{K-1} p(m) \prod_{t=0}^{T-1} P(s_{t+1} \mid s_t\,; m)\beta(a_t \mid s_t\,; m)R(r_t \mid s_t, s_{t+1}\,; m), \tag{4}$$

where $p(m)$ is the prior distribution on $m$. Modelling $P$, $R$, and $\beta$ with neural networks parametrised by $\theta$ results in a loss that depends on $\theta$. However, the multi-modality of (4) causes

Figure 2: Overview of our architecture. We also provided a diagram of the variBAD architecture in Figure 11 for comparison.

gradient-based optimisation algorithms to produce sub-optimal solutions. We circumvent this issue by introducing amortised inference network $q_\phi$ that takes a variable-length action-less trajectory $\tilde{\tau}_{:t} = (s_0, r_0, s_1, r_1, s_2, r_2, \ldots, s_{t-1}, r_{t-1}, s_t)$ as an input and outputs a distribution in $\mathcal{P}_d([K])$. Instead of maximising (4), we maximise the evidence lower bound (ELBO), which can be written by the following equation (Proposition 5):

$$\text{ELBO}_{\theta,\phi}(\tau\,;t) =$$

$$- D_{\text{KL}}(q_\phi(\tilde{\tau}_{:t}) \,\|\, p) + \sum_{i=0}^{T-1} \mathbb{E}_{m \sim q_\phi(\tilde{\tau}_{:t})} \left[ \log R_\theta(r_i \mid s_i, s_{i+1}\,;m) \right] \tag{5}$$

$$+ \sum_{i=0}^{T-1} \mathbb{E}_{m \sim q_\phi(\tilde{\tau}_{:t})} \left[ \log \beta_\theta(a_i \mid s_i\,;m) \right] + \sum_{i=0}^{T-1} \mathbb{E}_{m \sim q_\phi(\tilde{\tau}_{:t})} \left[ \log P_\theta(s_{i+1} \mid s_i\,;m) \right].$$

The first term $\log \rho_0(s_0)$ in (9) can be omitted because it is constant with respect to $\theta$ and $\phi$. The final objective for trajectory clustering is to maximise

$$\mathbb{E}_{\tau \sim \mathcal{D}} \left[ \frac{1}{T_\tau} \sum_{t=0}^{T_\tau - 1} \text{ELBO}_{\theta,\phi}(\tau\,;t) \right], \tag{6}$$

where $T_\tau$ is the length of the trajectory $\tau$ sampled from the dataset. An overview of our clustering algorithm is given in Figure 2.

After we finish training, we compute the behaviour policy estimations and cluster assignments according to the equations $\hat{\beta}_i = \beta_\theta(\cdot\,;i)$ and $\mathbb{A}(s) = \arg\max_m q_\phi(m \mid \tilde{\tau}(s))$, respectively, for each $i \in [K]$ and $s \in \mathcal{D}$, where $\tilde{\tau}(s)$ is the action-less trajectory containing the state $s$.

### 4.2 LOWER BOUND PENALISATION

In this section, we derive a new lower bound on the value function. As we did in the previous section, we start with the case where the offline dataset $\mathcal{D}$ is generated from a single behaviour policy $\beta$. The ideal penalisation method would be to use $Q^{\pi^*}(s, a)$ as a target, where $\pi^*$ is the optimal policy, but the value of $Q^{\pi^*}$ is inaccessible. So we aim to use a lower bound instead. In order to compute a lower bound, we first need to make some assumptions on the regularity of $P$ and $V^\beta$.

**Assumption 1.** There is $K_P > 0$ such that for all $s \in \mathcal{S}$ and $a, a' \in \mathcal{A}$, $W_1(P(s,a), P(s,a')) < K_P \|a - a'\|$, where $W_1(P, Q)$ is the Wasserstein distance of order 1 between two probability distributions $P, Q \in \mathcal{P}(\mathcal{S})$.

**Assumption 2.** The value function of the behaviour policy $\beta$ is $K_V$-Lipschitz, that is, for all $s, s' \in \mathcal{S}$, $\left| V^\beta(s) - V^\beta(s') \right| < K_V \|s - s'\|$.

Then, we can obtain a lower bound of $Q^{\pi^*}$ with these assumptions.

**Proposition 1.** *Define* $Q_\beta^{\text{LB}} \colon \mathcal{S} \times \mathcal{A} \to \mathbb{R}$ *by the equation*

$$Q_\beta^{\text{LB}}(s, a) = \max \left\{ V^\beta(s) - r_{\max} + r_{\min} - \gamma K_V K_P \, \mathbb{E}_{a' \sim \beta(s)} \left[ \|a - a'\| \right], \frac{r_{\min}}{1 - \gamma} \right\}. \tag{7}$$

*For any policy $\pi\colon \mathcal{S} \to \mathcal{P}(\mathcal{A})$ such that $V^{\pi} \succeq V^{\beta}$, $Q^{\pi} \succeq Q^{\mathrm{LB}}_{\beta}$.*

*Proof.* See page 15. □

Note that this lower bound is tighter than the previous bound $r_{\min}/(1-\gamma)$. The lower bound allows us to define the penalised Bellman optimality operator $\mathcal{T}^{\pi}_{\beta}$ for policy $\pi$ by the equation

$$(\mathcal{T}^{*}_{\beta}Q)(s,a) = \begin{cases} Q^{\mathrm{LB}}_{\beta}(s,a) & \text{if } a \in \mathrm{OOD}(s), \\ (\mathcal{T}^{*}Q)(s,a) & \text{otherwise}, \end{cases}$$

where $\mathcal{T}^{*}$ is the Bellman optimality operator defined as

$$(\mathcal{T}^{*}Q)(s,a) = \mathbb{E}_{s' \sim P(s,a), r \sim R(s,a,s')} \left[ r + \gamma \sup_{a' \in \mathcal{A}} Q(s',a') \right].$$

We can show that through repeated application of $\mathcal{T}^{*}_{\beta}$, it is possible to obtain a deterministic policy $\pi^{*}_{\beta}\colon \mathcal{S} \to \mathcal{A}$ that is optimal among the policies whose action for each state $s \in \mathcal{S}$ does not lie in $\mathrm{OOD}(s)$.

**Theorem 2.** *Any initial bounded real-valued function on $\mathcal{S} \times \mathcal{A}$ can converge to a unique fixed point $Q^{*}_{\beta}$ by repeatedly applying $\mathcal{T}^{*}_{\beta}$. Suppose for each $s \in \mathcal{S}$,*

$$Q^{\beta}(s,a_s) \geq \mathbb{E}_{a \sim \beta(s)} \left[ Q^{\beta}(s,a) \right]$$

*for some $a_s \in \mathcal{A} \setminus \mathrm{OOD}(s)$. If there exists a deterministic policy $\pi^{*}_{\beta}\colon \mathcal{S} \to \mathcal{A}$ that is optimal under the constraint $\pi(s) \notin \mathrm{OOD}(s)$ for all $s \in \mathcal{S}$, then $\pi^{*}_{\beta}(s) = \arg\max_{a \in \mathcal{A}} Q^{*}_{\beta}(s,a)$ for all $s \in \mathcal{S}$.*

*Proof.* See page 20. □

Now, the penalised Bellman optimality operator can easily be generalised to the heterogeneous dataset case with the set $\mathcal{B}$ of behaviour policy candidates and the set $\mathcal{V}(s)$ of valid behaviour policies given a state $s \in \mathcal{S}$.

$$(\mathcal{T}^{*}_{\mathcal{B}}Q)(s,a) = \begin{cases} Q^{\mathrm{LB}}_{\mathcal{B}}(s,a) & \text{if } a \in \mathrm{OOD}(s), \\ (\mathcal{T}^{*}Q)(s,a) & \text{otherwise}, \end{cases}$$

where $Q^{\mathrm{LB}}_{\mathcal{B}}\colon \mathcal{S} \times \mathcal{A} \to \mathbb{R}$ is defined as $Q^{\mathrm{LB}}_{\mathcal{B}}(s,a) = \max_{\beta \in \mathcal{V}(s)} Q^{\mathrm{LB}}_{\beta}(s,a)$ for each $s \in \mathcal{S}$ and $a \in \mathcal{A}$. We can prove a similar performance guarantee for the policy obtained by repeatedly applying $\mathcal{T}^{*}_{\mathcal{B}}$.

**Theorem 3.** *Any initial bounded real-valued function on $\mathcal{S} \times \mathcal{A}$ can converge to a unique fixed point $Q^{*}_{\mathcal{B}}$ by repeatedly applying $\mathcal{T}^{*}_{\mathcal{B}}$. Suppose for each $\beta \in \mathcal{B}$ and $s \in \mathcal{S}$,*

$$Q^{\beta}(s,a^{\beta}_s) \geq \mathbb{E}_{a \sim \beta(s)} \left[ Q^{\beta}(s,a) \right]$$

*for some $a^{\beta}_s \in \mathcal{A} \setminus \mathrm{OOD}(s)$. If there exists a deterministic policy $\pi^{*}_{\mathcal{B}}\colon \mathcal{S} \to \mathcal{A}$ that is optimal under the constraint $\pi(s) \notin \mathrm{OOD}(s)$ for all $s \in \mathcal{S}$, then $\pi^{*}_{\mathcal{B}}(s) = \arg\max_{a \in \mathcal{A}} Q^{*}_{\mathcal{B}}(s,a)$ for all $s \in \mathcal{S}$.*

*Proof.* See page 21. □

### 4.3 PRACTICAL ALGORITHM

The overall flow of our algorithm is as follows:

I. **Behaviour policy learning.** Run the trajectory clustering algorithm to obtain $\hat{\mathcal{B}}$ and $\hat{\mathcal{V}}(s)$. Or if it is known a priori that the dataset is homogeneous, then run a behaviour cloning algorithm to obtain $\hat{\beta}$.

II. **Behaviour value learning.** Learn a value function $\hat{V}^{\hat{\beta}}$ for each $\hat{\beta} \in \hat{\mathcal{B}}$ through temporal difference learning.

III. **Policy learning.** Obtain and apply $\mathcal{T}^{*}_{\mathcal{B}}$ repeatedly on a randomly initialised Q-function until convergence. Find a policy that maximises the learned Q-function.

This section mainly focuses on the trajectory clustering algorithm of Stage I. Additional details of our algorithm can be found in Section C of Appendix.

The network architecture used for trajectory clustering consists of three parts: the encoder, the latent sampler, and the decoder. The architecture is generally similar to that of variBAD except for a few adaptations. In this section, we will first go over how and why we modified each part. Then, we will propose a simple technique to adaptively set the number of clusters.

The encoder needs to take an action-less trajectory $\tilde{\tau}$ as an input and output the amortised posterior. Since the length of the trajectory may vary from one to another, the network should be capable of taking variable-length sequence as its input. For that purpose, variBAD utilises gated recurrent units (GRU; Cho et al. 2014). GRUs and other recurrent neural network variants suffer from the vanishing gradient problem (Bengio et al., 1994), which hampers their ability to process long sequences. Truncated backpropagation through time (Williams & Peng, 1990) can mitigate the phenomena to a certain extent, but we instead adopt the state space model architecture that is recently gaining interest in the area of sequence modelling (Gu et al., 2020; 2021; 2022; Gu & Dao, 2023; Dao & Gu, 2024). In particular, we use the S5 layer (Smith et al., 2023), which is simple and computationally efficient.

The second modification was made on the way latents are sampled and ELBOs are computed. As the latent variable in the variBAD architecture is continuous, it is impossible to analytically compute the expectation, and hence, the reparametrisation trick (Kingma & Welling, 2014) must be used. Although the latent variable is discrete in our case, exact computation is still inefficient because it requires multiple forward and backward passes through the decoder. We instead utilise the vector quantised-variational autoencoder (VQ-VAE; van den Oord et al. 2017) to approximate the ELBO. Under the VQ-VAE formulation, the amortised posterior $q_\phi$ is modelled as

$$q_\phi(m = k \mid \tilde{\tau}_{:t}) = \begin{cases} 1 & \text{if } k = k_\phi(\tilde{\tau}_{:t}) \\ 0 & \text{otherwise,} \end{cases}$$

where $e_0, e_1, \ldots, e_{K-1}$ are latent embedding vectors and

$$k_\phi(\tilde{\tau}_{:t}) = \arg\min_{j \in [K]} \|q_\phi(\tilde{\tau}_{:t}) - e_k\|_2.$$

Note that for simplicity, we have abused the notation $q_\phi$ to denote both the posterior and the encoder. The gradient flows into the encoder $q_\phi$ via the loss function

$$\ell_{\text{VQ}}(\phi; \tilde{\tau}_{:t}) = \left\| q_\phi(\tilde{\tau}_{:t}) - e_{k_\phi(\tilde{\tau}_{:t})} \right\|_2^2$$

and the latent embedding vectors are updated with exponential moving averages (EMA).

For the decoder, we use Gaussian distributions with diagonal covariance matrix to represent $P_\theta$, $\beta_\theta$, and $R_\theta$. Most RL environments have a bounded action space, whereas a Gaussian distribution has unbounded support. To estimate the behaviour policy more accurately, we first normalize the actions between $-1$ and 1 and apply the inverse hyperbolic tangent function on each dimension of the actions to map them onto $\mathbb{R}^{d_a}$. Note that we use the mapped actions when learning the critic, that is, the critic function takes $\tanh^{-1}(a)$ instead of $a$ as input. Finally, instead of taking the summation over the entire trajectory in (5) and (6), we adopt the implementation trick of variBAD and randomly subsample $N_d$ transition steps in (5) and $N_e$ ELBO terms in (6). To conclude, the loss function for the trajectory clustering algorithm is

$$\ell_{\text{TC}}(\theta, \phi; \tau) = \frac{1}{N_e N_d} \sum_{t \in \mathcal{I}_e} \sum_{i \in \mathcal{I}_d} A_\theta(s_i, a_i, r_i, s_{i+1}; e_{k_\phi(\tilde{\tau}_{:t})}) + \lambda_{\text{VQ}} \ell_{\text{VQ}}(\phi; \tilde{\tau}_{:t}),$$

where

$$A_\theta(s, a, r, s'; m) = \log \beta_\theta(a \mid s; m) + \lambda_T \log P_\theta(s' \mid s; m) + \lambda_R \log R_\theta(r \mid s, s'; m), \quad (8)$$

$\mathcal{I}_e$ and $\mathcal{I}_d$ are sets of indices sampled uniformly at random with replacement from $[T_\tau]$ with sizes $N_e$ and $N_d$, respectively, and $\lambda_{\text{VQ}}, \lambda_T, \lambda_R$ are tunable hyperparameters.

Choosing the right number of clusters is crucial for high performance in most clustering algorithms. To alleviate the burden of hyperparameter tuning, we adopt a two-phase training paradigm. During

Table 2: Average normalised scores on the D4RL benchmark. Note that "ha" means halfcheetah, "ho" means hopper, "wa" means walker2d, "m" means medium, "r" means replay, "ra" means random, and "e" means expert.

| Dataset | BC | TD3BC | BCQ | BEAR | CQL | IQL | MCQ | SVR | Ours |
|---------|------|-------|-------|------|-------|-------|-------|-------|----------------|
| ha-ra | 2.6 | 11.0 | 2.2 | 2.3 | 17.5 | 13.1 | 28.5 | 27.2 | $27.0 \pm 1.1$ |
| ho-ra | 4.1 | 8.5 | 7.8 | 3.9 | 7.9 | 7.9 | 31.8 | 31.0 | $31.5 \pm 0.2$ |
| wa-ra | 1.2 | 1.6 | 4.9 | 12.8 | 5.1 | 5.4 | 17.0 | 2.2 | $16.6 \pm 7.9$ |
| ha-m | 42.0 | 48.3 | 46.6 | 43.0 | 47.0 | 47.4 | 64.3 | 60.5 | $63.5 \pm 1.2$ |
| ho-m | 56.2 | 59.3 | 59.4 | 51.8 | 53.0 | 66.2 | 78.4 | 103.5 | $103.4 \pm 0.9$ |
| wa-m | 71.0 | 83.7 | 71.8 | $-0.2$ | 73.3 | 78.3 | 91.0 | 92.4 | $96.5 \pm 13.9$ |
| ha-m-r | 36.4 | 44.6 | 42.2 | 36.3 | 45.5 | 44.2 | 56.8 | 52.5 | $52.2 \pm 0.8$ |
| ho-m-r | 21.8 | 60.9 | 60.9 | 52.2 | 88.7 | 94.7 | 101.6 | 103.7 | $102.2 \pm 1.1$ |
| wa-m-r | 24.9 | 81.8 | 57.0 | 7.0 | 81.8 | 73.8 | 91.3 | 95.6 | $95.4 \pm 19.2$ |
| ha-m-e | 59.6 | 90.7 | 95.4 | 46.0 | 75.6 | 86.7 | 87.5 | 94.2 | $90.9 \pm 4.2$ |
| ho-m-e | 51.7 | 98.0 | 106.9 | 50.6 | 105.6 | 91.5 | 111.2 | 111.2 | $112.4 \pm 1.1$ |
| wa-m-e | 101.2 | 110.1 | 107.7 | 22.1 | 107.9 | 109.6 | 114.2 | 109.3 | $108.3 \pm 0.7$ |
| ha-e | 88.2 | 81.7 | 92.7 | 92.9 | 96.3 | 95.0 | 96.2 | 96.1 | $96.6 \pm 0.9$ |
| ho-e | 110.9 | 107.8 | 109.0 | 54.6 | 96.5 | 109.4 | 111.4 | 111.1 | $112.7 \pm 0.9$ |
| wa-e | 107.7 | 110.2 | 106.3 | 106.6 | 108.5 | 109.9 | 107.2 | 110.0 | $113.4 \pm 0.5$ |
| Average | 52.3 | 67.5 | 64.5 | 38.8 | 67.3 | 68.9 | 79.2 | 80.0 | 81.5 |

the first phase of the paradigm, we set the codebook size to be sufficiently large. After completing the first phase, we compute the cluster assignments for each state in the dataset. If the number of states assigned to a particular cluster does not exceed a certain threshold, we remove the corresponding code from the VQ-VAE codebook. The training is resumed with the remaining codebook. This way, we could adaptively determine the number of clusters without needing to perform an exhaustive hyperparameter search.

## 5 EXPERIMENTS

### 5.1 RESULTS ON THE D4RL BENCHMARK

In order to evaluate how well our algorithm perform on various offline RL tasks, we tested our method on the D4RL (Fu et al., 2020) benchmark. We compared it with existing offline RL methods such as BC (Pomerleau, 1988), TD3+BC (Fujimoto & Gu, 2021), BCQ (Fujimoto et al., 2019), CQL (Kumar et al., 2020), BEAR (Kumar et al., 2019), IQL (Kostrikov et al., 2022), MCQ (Lyu et al., 2022), and SVR (Mao et al., 2023). We trained our method with five different seeds to obtain five different policies and sampled ten trajectories with each of them. We report the average and standard deviation of the fifty normalized scores in Table 2. The results show that our algorithm can successfully learn high-performing policies from most datasets, while attaining state-of-the-art scores on some of them.

### 5.2 EXPERIMENTS ON A HETEROGENEOUS DATASET

Although D4RL datasets such as "hopper-medium-expert-v2" were sampled with more than one behaviour policies, the action distributions are actually unimodal on most states due to the state distribution being so different between the trajectories of the two behaviour policies. Figure 3 presents a visualisation of the entire and initial state distributions of the "hopper-medium-expert-v2" dataset where we have used the uniform manifold approximation and projection (UMAP; McInnes & Healy 2018) technique for dimension reduction. We can see that expert and medium states are clearly separated, except for the initial states.

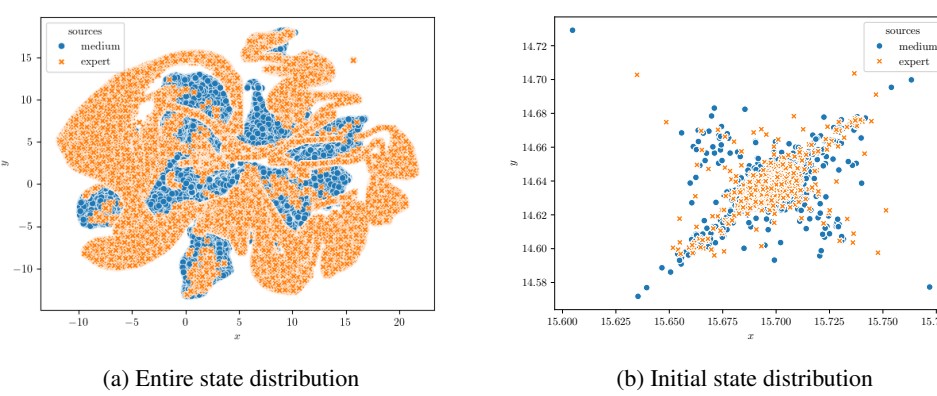

(a) Entire state distribution

(b) Initial state distribution

Figure 3: The UMAP of the states in the "hopper-medium-expert-v2" dataset.

To demonstrate the effectiveness of our trajectory clustering algorithm, we created a custom dataset with drastically different initial state behaviours using the "Hopper-v5" environment provided by the Gymnasium library (Towers et al., 2024). Half of the samples in the dataset were sampled from an expert policy, and the other half was sampled from a policy that tripped over within eight timesteps. Table 3 demonstrates that our method can effectively classify the two datasets and learn an optimal policy from a truly heterogeneous dataset.

Table 3: The performance of SVR and our method on the custom heterogeneous dataset.

| Algorithm | Length | Return |
|---|---|---|
| SVR | $8.00 \pm 0.00$ | $4.05 \pm 0.01$ |
| Ours | $436.3 \pm 32.1$ | $4062.0 \pm 24.5$ |

### 5.3 ANALYSIS ON THE TRAJECTORY CLUSTERING ALGORITHM

In meta reinforcement learning settings, each MDP has independent transition and reward dynamics, so they must be modelled in order to infer the MDP from trajectories. Under our formulation, on the other hand, transition and reward dynamics of each MRP are correlated with each other through the policy as we can see from (2) and (3). Although this implies that we may identify the MRP solely through modelling the behaviour policy, we hypothesized that modelling transition and reward dynamics can provide meaningful auxiliary information leading to better clustering performance. Therefore, we compared the performance of our algorithm under four different configurations $(\lambda_T, \lambda_R) \in \{(1,1), (1,0), (0,1), (0,0)\}$, where $\lambda_T$ and $\lambda_R$ are the weights for transition

Table 4: The impact of hyperparameters $\lambda_T$ and $\lambda_R$ on the average performance of our trajectory clustering algorithm evaluated on six custom D4RL datasets. The performance is measured in terms of adjusted rand index (ARI) and normalized mutual information score (NMI).

| $\lambda_R$ | $\lambda_T$ | ARI | NMI |
|---|---|---|---|
| 0 | 0 | $0.98 \pm 0.07$ | $0.98 \pm 0.06$ |
| 0 | 1 | $0.91 \pm 0.21$ | $0.92 \pm 0.17$ |
| 1 | 0 | $0.99 \pm 0.02$ | $0.98 \pm 0.02$ |
| 1 | 1 | $0.86 \pm 0.27$ | $0.87 \pm 0.24$ |

and reward models defined in (8). To evaluate the accuracy of our trajectory clustering algorithm, we created custom D4RL datasets by concatenating random, medium, and expert datasets. The mean and standard deviation of adjusted rand indices (ARI; Hubert & Arabie 1985) and normalised mutual information scores (NMI) for each configuration over 5 different seeds are reported in Table 4. We can see that the configuration $(\lambda_T, \lambda_R) = (1, 0)$ performs the best on average. Unlike $s_{i+1}$, which is in the vicinity of $s_i$ regardless of the $a_i$, $r_i$ can vary drastically between policies, making it difficult to model rewards from different policies with a single neural network. We speculate this to be the reason why training a reward model negatively affects the performance of our trajectory clustering algorithm. For experiments on other datasets, refer to Section D.2.

Table 5: The impact of the hyperparamter $\alpha$ on the average normalised score on three different datasets in the D4RL benchmark. Note that "ha" means halfcheetah, "ho" means hopper, "wa" means walker2d, and "m" means medium.

| Dataset | $\alpha = 0.1$ | $\alpha = 0.3$ | $\alpha = 0.5$ | $\alpha = 0.7$ | $\alpha = 0.9$ |
|---------|---------|---------|---------|---------|---------|
| ha-m | $63.2 \pm 1.1$ | $62.4 \pm 1.4$ | $63.5 \pm 1.2$ | $63.3 \pm 1.0$ | $62.5 \pm 1.2$ |
| ho-m | $102.2 \pm 0.5$ | $103.3 \pm 0.9$ | $103.4 \pm 0.9$ | $102.4 \pm 0.7$ | $103.1 \pm 0.5$ |
| wa-m | $93.0 \pm 22.9$ | $91.3 \pm 22.4$ | $96.1 \pm 13.9$ | $94.2 \pm 20.4$ | $97.0 \pm 15.2$ |

Table 6: The impact of the hyperparamter $K$ on the average normalised score on three different datasets in the D4RL benchmark. Note that "ha" means halfcheetah, "ho" means hopper, "wa" means walker2d, and "m" means medium.

| Dataset | $K = 0.1$ | $K = 0.2$ | $K = 0.5$ | $K = 1.0$ | $K = 2.0$ |
|---------|---------|---------|---------|---------|---------|
| ha-m | $62.5 \pm 0.7$ | $63.1 \pm 1.4$ | $63.5 \pm 1.2$ | $63.4 \pm 0.9$ | $63.1 \pm 1.1$ |
| ho-m | $102.1 \pm 2.8$ | $102.6 \pm 0.7$ | $102.2 \pm 0.8$ | $102.0 \pm 5.7$ | $103.4 \pm 0.9$ |
| wa-m | $86.9 \pm 27.6$ | $90.6 \pm 25.0$ | $93.1 \pm 26.7$ | $96.1 \pm 13.9$ | $93.9 \pm 15.6$ |

| Dataset | $K = 5.0$ | $K = 10.0$ | $K = 20.0$ | $K = 50.0$ | $K = 100.0$ |
|---------|---------|---------|---------|---------|---------|
| ha-m | $63.3 \pm 1.3$ | $62.8 \pm 0.8$ | $63.2 \pm 1.2$ | $62.6 \pm 1.0$ | $62.1 \pm 1.2$ |
| ho-m | $102.3 \pm 3.9$ | $102.5 \pm 1.5$ | $100.0 \pm 6.8$ | $84.1 \pm 20.7$ | $92.5 \pm 21.1$ |
| wa-m | $92.3 \pm 1.6$ | $89.5 \pm 1.5$ | $88.0 \pm 1.6$ | $81.7 \pm 9.1$ | $78.4 \pm 13.7$ |

## 5.4 ABLATION STUDY

We investigate the impact of the choice of hyperparameters $\alpha$ and $K = K_V K_P$ on the performance of our method on three different datasets: halfcheetah-medium-v2, hopper-medium-v2, and walker2d-medium-v2. As shown in Table 5, the performance is robust to a wide range of $\alpha$ values. Similarly, we can see from Table 6 that the performance remains stable for moderate choices of $K$, While large values ($K \geq 50$) lead to degradation, particularly for hopper-medium-v2. Overall, the result indicate that our method does not require precise tuning on $\alpha$ and $K$ to achieve strong performance.

## 6 CONCLUSION

In this paper, we propose a new value regularisation algorithm for offline RL penalizing their critic values, based on the OOD action set that we were able to explicitly identify. We determine how OOD an action is based on its likelihood, where the threshold is set adaptively according to the shape of the behaviour policy. To enable likelihood analysis for heterogeneous datasets where simple behaviour cloning fails, we introduce a novel trajectory clustering technique based on a meta-learning formulation of the clustering problem. Our method of penalising the critic values for OOD actions by regressing them towards a lower bound of the optimal Q-value function is proven to be effective both theoretically and empirically.

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

## A NOTATIONS

- $\mathbf{0}$: a zero vector with dimensionality implied by context
- $D_{\mathrm{KL}}(P_1 \parallel P_2)$: the Kullback–Leibler (KL) divergence from a probability distribution $P_1$ to another probability distribution $P_2$
- $\mathbf{e}_i$: the $i$-th standard basis of a Euclidean space
- $f(y \mid x)$: the value of the pdf (or pmf) of the distribution $f(x)$ at $y$, where $Y$ is a set and $f\colon X \to \mathcal{P}(Y)$
- $f \succeq g$: $f(x) \geq g(x)$ for all $x \in X$, where $f$ and $g$ are real-valued functions defined on a set $X$
- $f \equiv g$: $f(x) = g(x)$ for all $x \in X$, where $f$ and $g$ are real-valued functions defined on a set $X$
- $\boldsymbol{I}_d$: an identity matrix with $d$ rows and $d$ columns
- $L^{\infty}(X)$: the space of bounded real value functions on a set $X$ endowed with the supremum norm
- $[N]$: the set $\{0, 1, \ldots, N-1\}$, where $N$ is an integer
- $\mathcal{N}(\boldsymbol{\mu}, \boldsymbol{\Sigma})$: a multi-variate Gaussian distribution with mean vector $\boldsymbol{\mu}$ and covariance matrix $\boldsymbol{\Sigma}$
- $\mathbb{P}(E)$: probability of an event $E$
- $\mathcal{P}(X)$: family of absolutely continuous probability distributions with finite first moments supported on a subset of $X$, where $X \subseteq \mathbb{R}^d$
- $\mathcal{P}_d(X)$: the family of discrete distributions supported on a subset of $X$, where $X \subseteq \mathbb{R}^d$

- supp $\mu$: the support of a probability distribution $\mu$
- $W_1(P_1, P_2)$: the Wasserstein distance of order 1 between two probability distributions $P_1, P_2 \in \mathcal{P}(X)$

# B  PROOFS

**Proposition 4.** *Let $\mathbf{X}$ be a multivariate Gaussian random variable with mean vector $\boldsymbol{\mu} \in \mathbb{R}^d$ and positive definite covariance matrix $\boldsymbol{\Sigma} \in \mathbb{R}^{d \times d}$. The $100(1-\alpha)\%$ HDR is*

$$\mathcal{R}(f_\alpha) = \left\{ \mathbf{x} \in \mathbb{R}^d : (\mathbf{x} - \boldsymbol{\mu})^\top \boldsymbol{\Sigma}^{-1} (\mathbf{x} - \boldsymbol{\mu}) \leq F_{\chi_d^2}^{-1}(1 - \alpha) \right\},$$

*where $F_{\chi_d^2}$ is the cumulative distribution function of a chi-squared random variable with $d$ degrees of freedom.*

*Proof.* Let $\mathbf{Z} = (Z_1, Z_2, \ldots, Z_d) = \sqrt{\boldsymbol{\Sigma}^{-1}}(\mathbf{X} - \boldsymbol{\mu})$. By the change of variables formula,

$$p_{\mathbf{Z}}(\mathbf{z}) = \left| \det(\sqrt{\boldsymbol{\Sigma}}) \right| p_{\mathbf{X}}\left( \boldsymbol{\mu} + \sqrt{\boldsymbol{\Sigma}} \mathbf{z} \right)$$

$$= \det(\boldsymbol{\Sigma})^{1/2}(2\pi)^{-d/2} \det(\boldsymbol{\Sigma})^{-1/2} \exp\left( -\frac{1}{2} \mathbf{z}^\top \mathbf{z} \right)$$

$$= (2\pi)^{-d/2} \exp\left( -\frac{1}{2} \mathbf{z}^\top \mathbf{z} \right),$$

where $p_{\mathbf{X}}$ and $p_{\mathbf{Z}}$ are the pdfs of random vectors $\mathbf{X}$ and $\mathbf{Z}$, respectively. We can see that $\mathbf{Z}$ is a standard normal random vector. Since

$$\mathcal{R}(y) = \left\{ \mathbf{x} \in \mathbb{R}^d : (2\pi)^{-d/2} \det(\boldsymbol{\Sigma})^{-1/2} \exp\left( -\frac{1}{2}(\mathbf{x} - \boldsymbol{\mu})^\top \boldsymbol{\Sigma}^{-1}(\mathbf{x} - \boldsymbol{\mu}) \right) \geq y \right\}$$

$$= \left\{ \mathbf{x} \in \mathbb{R}^d : (\mathbf{x} - \boldsymbol{\mu})^\top \boldsymbol{\Sigma}^{-1}(\mathbf{x} - \boldsymbol{\mu}) \leq -2 \log y + d \log(2\pi) + \log \det(\Sigma) \right\},$$

we have

$$\mathbb{P}(\mathbf{X} \in \mathcal{R}(y)) = \mathbb{P}\left( \mathbf{Z}^\top \mathbf{Z} \leq -2 \log y + d \log(2\pi) + \log \det(\Sigma) \right)$$

$$= \mathbb{P}\left( \sum_{i=1}^d Z_i^2 \leq -2 \log y + d \log(2\pi) + \log \det(\Sigma) \right).$$

$Z_1, Z_2, \ldots, Z_d$ are independent, so $\sum_{i=1}^d Z_i^2$ is a chi-squared random variable. This implies

$$\mathbb{P}(\mathbf{X} \in \mathcal{R}(y)) = F_{\chi_d^2}(-2 \log y + d \log(2\pi) + \log \det(\boldsymbol{\Sigma})).$$

$\mathbb{P}(\mathbf{X} \in \mathcal{R}(y)) \geq 1 - \alpha$ if and only if

$$-2 \log y + d \log(2\pi) + \log \det(\boldsymbol{\Sigma}) \geq F_{\chi_d^2}^{-1}(1 - \alpha).$$

Therefore,

$$f_\alpha = (2\pi)^{d/2} \det(\boldsymbol{\Sigma})^{1/2} \exp\left( -\frac{1}{2} F_{\chi_d^2}(1 - \alpha) \right),$$

which means

$$\mathcal{R}(f_\alpha) = \left\{ \mathbf{x} \in \mathbb{R}^d : (\mathbf{x} - \boldsymbol{\mu})^\top \boldsymbol{\Sigma}^{-1}(\mathbf{x} - \boldsymbol{\mu}) \leq F_{\chi_d^2}^{-1}(1 - \alpha) \right\}.$$

$\square$

**Proposition 5.** *Let $m$ be a discrete latent variable supported on $[K]$ and*

$$\tau_{:T} = (s_0, a_0, r_0, s_1, a_1, r_1, s_2, a_2, r_2, \ldots, s_{T-1}, a_{T-1}, r_{T-1}, s_T)$$

*be a trajectory rolled-out according to the following sampling process: $s_0 \sim \rho_0$, $m \sim p$, and for each $t \in [T]$, $s_{t+1} \sim P(s_t; m)$, $a_t \sim \beta(s_t; m)$, and $r_t \sim R(s_t, s_{t+1}; m)$. The marginal pdf can be written as*

$$p(\tau_{:T}) = \rho_0(s_0) \sum_{m=0}^{K-1} p(m) \prod_{t=0}^{T-1} P(s_{t+1} \mid s_t; m) \beta(a_t \mid s_t; m) R(r_t \mid s_t, s_{t+1}; m)$$

*and for any distribution $q$ on $[K]$,*

$$\log p(\tau_{:T}) \geq \log \rho_0(s_0) - D_{\text{KL}}(q \parallel p) \tag{9}$$

$$+ \sum_{t=0}^{T-1} \mathbb{E}_{m \sim q} \left[ \log P(s_{t+1} \mid s_t \, ; m) + \log \beta(a_t \mid s_t \, ; m) + \log R(r_t \mid s_t, s_{t+1} \, ; m) \right].$$

*Proof.* Let us denote the action-less trajectory by $\tilde{\tau}_{:T}$, that is,

$$\tilde{\tau}_{:T} = (s_0, r_0, s_1, r_1, s_2, r_2, \ldots, s_{T-1}, r_{T-1}, s_T).$$

By Jensen's inequality,

$$\log p(\tau_{:T})$$

$$= \log \rho_0(s_0) + \log \sum_{m=0}^{K-1} p(m) \prod_{t=0}^{T-1} \left[ P(s_{t+1} \mid s_t \, ; m) \beta(a_t \mid s_t \, ; m) R(r_t \mid s_t, s_{t+1} \, ; m) \right]$$

$$= \log \rho_0(s_0) + \log \sum_{m=0}^{K-1} q(m) \cdot \frac{p(m)}{q(m)} \prod_{t=0}^{T-1} \left[ P(s_{t+1} \mid s_t \, ; m) \beta(a_t \mid s_t \, ; m) R(r_t \mid s_t, s_{t+1} \, ; m) \right]$$

$$\geq \log \rho_0(s_0) + \mathbb{E}_{m \sim q} \left[ \log \frac{p(m)}{q(m)} + \sum_{t=0}^{T-1} A(s_t, a_t, r_t, s_{t+1} \, ; m) \right]$$

$$= \log \rho_0(s_0) - D_{\text{KL}}(q \parallel p) + \sum_{t=0}^{T-1} \mathbb{E}_{m \sim q} \left[ A(s_t, a_t, r_t, s_{t+1} \, ; m) \right],$$

where

$$A(s_t, a_t, r_t, s_{t+1} \, ; m) = \log P(s_{t+1} \mid s_t \, ; m) + \log \beta(a_t \mid s_t \, ; m) + \log R(r_t \mid s_t, s_{t+1} \, ; m).$$

$\square$

We restate the two assumptions we made in Section 4.2 for the reader's convenience.

**Assumption 3.** There is $K_P > 0$ such that for all $s \in \mathcal{S}$ and $a_1, a_2 \in \mathcal{A}$, $W_1(P(s, a_1), P(s, a_2)) < K_P \|a_1 - a_2\|$.

**Assumption 4.** The value function of the behaviour policy $\beta$ is $K_V$-Lipschitz.

**Lemma 6.** *For any policy $\pi$ and $s \in \mathcal{S}$,*

$$\frac{r_{\min}}{1 - \gamma} \leq V^\beta(s) \leq \frac{r_{\max}}{1 - \gamma}.$$

*Proof.* By the definition of $V^\pi$, for all $s \in \mathcal{S}$,

$$V^\pi(s) = \mathbb{E}_{\tau \sim \pi \mid s} \left[ \sum_{t=0}^\infty \gamma^t r_t \right] \geq \mathbb{E}_{\tau \sim \pi \mid s} \left[ \sum_{t=0}^\infty \gamma^t r_{\min} \right] = \frac{r_{\min}}{1 - \gamma},$$

and

$$V^\pi(s) = \mathbb{E}_{\tau \sim \pi \mid s} \left[ \sum_{t=0}^\infty \gamma^t r_t \right] \leq \mathbb{E}_{\tau \sim \pi \mid s} \left[ \sum_{t=0}^\infty \gamma^t r_{\max} \right] = \frac{r_{\max}}{1 - \gamma}.$$

$\square$

**Proposition 7.** *Define $Q_\beta^{\text{LB}} \colon \mathcal{S} \times \mathcal{A} \to \mathbb{R}$ by the equation*

$$Q_\beta^{\text{LB}}(s, a) = \max \left\{ V^\beta(s) - r_{\max} + r_{\min} - \gamma K_V K_P \, \mathbb{E}_{a' \sim \beta(s)} \left[ \|a - a'\| \right], \frac{r_{\min}}{1 - \gamma} \right\}.$$

*For any policy $\pi \colon \mathcal{S} \to \mathcal{P}(\mathcal{A})$ such that $V^\pi \succeq V^\beta$, $Q^\pi \succeq Q_\beta^{\text{LB}}$.*

*Proof.* By Lemma 6 and the definition of $Q^\pi$, for all $s \in \mathcal{S}$ and $a \in \mathcal{A}$,

$$Q^\pi(s, a) = \mathbb{E}_{s' \sim P(s,a,s'), r \sim R(s,a,s')} \left[ r + \gamma V^\pi(s') \right]$$

$$\geq \mathbb{E}_{s' \sim P(s,a,s'), r \sim R(s,a,s')} \left[ r_{\min} + \gamma \frac{r_{\min}}{1 - \gamma} \right]$$

$$= \frac{r_{\min}}{1 - \gamma}.$$

So we only need to show that for all $s \in \mathcal{S}$ and $a \in \mathcal{A}$,

$$Q^\pi(s, a) \geq V^\beta(s) - r_{\max} + r_{\min} - \gamma K_V K_P \, \mathbb{E}_{a' \sim \beta(s)} \left[ \|a - a'\| \right].$$

Let $a_1, a_2 \in \mathcal{A}$. By the Kantorovich–Rubinstein formula (Villani, 2009),

$$\left| \mathbb{E}_{s' \sim P(\cdot|s,a_1)}[V^\beta(s')] - \mathbb{E}_{s' \sim P(\cdot|s,a_2)}[V^\beta(s')] \right| \leq K_V W_1(P(\cdot \mid s, a_1), P(\cdot \mid s, a_2))$$

$$\leq K_V K_P \|a_1 - a_2\|.$$

Therefore,

$$Q^\pi(s, a) = \mathbb{E}_{s' \sim P(s,a), r \sim R(s,a,s')} \left[ r + \gamma V^\pi(s') \right]$$

$$\geq \mathbb{E}_{s' \sim P(s,a), r \sim R(s,a,s')} \left[ r + \gamma V^\beta(s') \right]$$

$$= V^\beta(s) - \mathbb{E}_{a' \sim \beta(\cdot|s)} \left[ \mathbb{E}_{s' \sim P(s,a'), r \sim R(s,a',s')} \left[ r + \gamma V^\beta(s') \right] \right]$$

$$\quad + \mathbb{E}_{s' \sim P(s,a), r \sim R(s,a,s')} \left[ r + \gamma V^\beta(s') \right]$$

$$\geq V^\beta(s) - r_{\max} + r_{\min} + \gamma \, \mathbb{E}_{a' \sim \beta(s)} \left[ \mathbb{E}_{s' \sim P(s,a)} \left[ V^\beta(s') \right] - \mathbb{E}_{s' \sim P(s,a')} \left[ V^\beta(s') \right] \right]$$

$$\geq V^\beta(s) - r_{\max} + r_{\min} - \gamma K_V K_P \, \mathbb{E}_{a' \sim \beta(\cdot|s)} \left[ \|a - a'\| \right].$$

Note that we have used the fact that

$$V^\beta(s) = \mathbb{E}_{a' \sim \beta(s), s' \sim P(s,a'), r \sim R(s,a',s')} \left[ r + \gamma V^\beta(s') \right].$$

$\square$

**Theorem 8.** *Let $\{A_s\}_{s \in \mathcal{S}}$ be a family of subsets of $\mathcal{A}$, $\tilde{Q} \in L^\infty(\mathcal{S} \times \mathcal{A})$, and $\mathcal{T}_A$ be an operator on the space of real-valued functions on $\mathcal{S} \times \mathcal{A}$ defined by the equation*

$$(\mathcal{T}_A Q)(s, a) = \begin{cases} (\mathcal{T}^* Q)(s, a) & \text{if } a \in A_s, \\ \tilde{Q}(s, a) & \text{otherwise,} \end{cases}$$

*for each $Q \in L^\infty(\mathcal{S} \times \mathcal{A})$. Then any bounded real-valued function on $\mathcal{S} \times \mathcal{A}$ converges to a unique fixed point $Q_A$ by repeatedly applying $\mathcal{T}_A$.*

*Proof.* Fix $s \in \mathcal{S}$, $a \in \mathcal{A}$, and $Q \in L^\infty(\mathcal{S} \times \mathcal{A})$. If $a \in A_s$,

$$|(\mathcal{T}_A Q)(s, a)| = |(\mathcal{T}^* Q)(s, a)|$$

$$= \left| \mathbb{E}_{s' \sim P(s,a), r \sim R(s,a,s')} \left[ r + \gamma \sup_{a' \in \mathcal{A}} Q(s', a') \right] \right|$$

$$\leq \mathbb{E}_{s' \sim P(s,a), r \sim R(s,a,s')} \left[ \left| r + \gamma \sup_{a' \in \mathcal{A}} Q(s', a') \right| \right]$$

$$\leq \mathbb{E}_{s' \sim P(s,a), r \sim R(s,a,s')} \left[ |r| + \gamma \left| \sup_{a' \in \mathcal{A}} Q(s', a') \right| \right]$$

$$\leq \mathbb{E}_{s' \sim P(s,a), r \sim R(s,a,s')} \left[ \max \left\{ |r_{\max}|, |r_{\min}| \right\} + \gamma \|Q\|_\infty \right],$$

$$= \max \left\{ |r_{\max}|, |r_{\min}| \right\} + \gamma \|Q\|_\infty.$$

Otherwise,

$$|(\mathcal{T}_A Q)(s, a)| = \left| \tilde{Q}(s, a) \right| \leq \|\tilde{Q}\|_\infty.$$

So

$$\|\mathcal{T}_A Q\|_\infty \leq \max \left\{ \|\tilde{Q}\|_\infty, \max \left\{ |r_{\max}|, |r_{\min}| \right\} + \gamma \|Q\|_\infty \right\} < \infty,$$

that is, $\mathcal{T}_A Q \in L^\infty(\mathcal{S} \times \mathcal{A})$. So the restriction of $\mathcal{T}_A$ onto $L^\infty(\mathcal{S} \times \mathcal{A})$ is an operator on $L^\infty(\mathcal{S} \times \mathcal{A})$. With a slight abuse of notation, we will just denote the restriction by $\mathcal{T}_A$ from now on.

Now we go on and prove that $\mathcal{T}_A$ is a contraction operator. Fix $s \in \mathcal{S}$, $a \in \mathcal{A}$ and $Q_1, Q_2 \in L^\infty(\mathcal{S} \times \mathcal{A})$. If $a \in A_s$,

$$
\begin{aligned}
|(\mathcal{T}_A Q_1)(s,a) - (\mathcal{T}_A Q_2)(s,a)| &= |(\mathcal{T}^* Q_1)(s,a) - (\mathcal{T}^* Q_2)(s,a)| \\
&= \gamma \left| \mathbb{E}_{s' \sim P(s,a)} \left[ \sup_{a' \in \mathcal{A}} Q_1(s',a') - \sup_{a'' \in \mathcal{A}} Q_2(s',a'') \right] \right| \\
&\leq \gamma \, \mathbb{E}_{s' \sim P(s,a)} \left[ \left| \sup_{a' \in \mathcal{A}} Q_1(s',a') - \sup_{a'' \in \mathcal{A}} Q_2(s',a'') \right| \right] \\
&\leq \gamma \, \mathbb{E}_{s' \sim P(s,a)} \left[ \sup_{a' \in \mathcal{A}} |Q_1(s',a') - Q_2(s',a')| \right] \\
&\leq \gamma \|Q_1 - Q_2\|_\infty.
\end{aligned}
$$

Otherwise,

$$
|(\mathcal{T}_A Q_1)(s,a) - (\mathcal{T}_A Q_2)(s,a)| = \left| \tilde{Q}(s,a) - \tilde{Q}(s,a) \right| = 0 \leq \gamma \|Q_1 - Q_2\|_\infty.
$$

Therefore, and $\mathcal{T}_A$ is a contraction mapping on $L^\infty(\mathcal{S} \times \mathcal{A})$. By the contraction mapping theorem, any initial-bounded Q-function would converge to a unique fixed point $Q_A$. $\qquad\square$

**Lemma 9.** *Let $\pi_1$ and $\pi_2$ be two policies. If $\mathbb{E}_{a \sim \pi_1(s)}[Q^{\pi_2}(s,a)] \geq V^{\pi_2}(s)$ for all $s \in \mathcal{S}$, then $V^{\pi_1} \succeq V^{\pi_2}$.*

*Proof.* We define a sequence $(Q_n)$ of bounded real-valued functions on $\mathcal{S} \times \mathcal{A}$ by the recurrence relation

$$
Q_n = \begin{cases} Q^{\pi_2} & \text{if } n = 0, \\ \mathcal{T}^{\pi_1} Q_{n-1} & \text{otherwise.} \end{cases}
$$

We first show that $Q_n \succeq Q^{\pi_2}$ by mathematical induction. The base case is trivial because $Q_0 \equiv Q^{\pi_2}$. Suppose $Q_{n-1} \succeq Q^{\pi_2}$. Then for each $s \in \mathcal{S}$ and $a \in \mathcal{A}$,

$$
\begin{aligned}
Q_n(s,a) &= (\mathcal{T}^{\pi_1} Q_{n-1})(s,a) \\
&= \mathbb{E}_{s' \sim P(s,a), r \sim R(s,a,s')} \left[ r + \gamma \, \mathbb{E}_{a' \sim \pi_1(s')} [Q_{n-1}(s',a')] \right] \\
&\geq \mathbb{E}_{s' \sim P(s,a), r \sim R(s,a,s')} \left[ r + \gamma \, \mathbb{E}_{a' \sim \pi_1(s')} [Q^{\pi_2}(s',a')] \right] \\
&\geq \mathbb{E}_{s' \sim P(s,a), r \sim R(s,a,s')} \left[ r + \gamma \, \mathbb{E}_{a' \sim \pi_2(s')} [Q^{\pi_2}(s',a')] \right] \\
&= (\mathcal{T}^{\pi_2} Q^{\pi_2})(s,a) \\
&= Q^{\pi_2}(s,a).
\end{aligned}
$$

So $Q_n \succeq Q^{\pi_2}$. By mathematical induction, $Q_n \succeq Q^{\pi_2}$ for all $n$. For all $s \in \mathcal{S}$ and $a \in \mathcal{A}$,

$$
Q^{\pi_1}(s,a) = \lim_{n \to \infty} Q_n(s,a) \geq Q^{\pi_2}(s,a).
$$

Therefore, for all $s \in \mathcal{S}$,

$$
V^{\pi_1}(s) = \mathbb{E}_{a \sim \pi_1(s)} [Q^{\pi_1}(s,a)] \geq \mathbb{E}_{a \sim \pi_1(s)} [Q^{\pi_2}(s,a)] \geq V^{\pi_2}(s),
$$

that is, $V^{\pi_1} \succeq V^{\pi_2}$. $\qquad\square$

**Theorem 10.** *Let $\{A_s\}_{s \in \mathcal{S}}$ be a family of subsets of $\mathcal{A}$, $\tilde{Q} \in L^\infty(\mathcal{S} \times \mathcal{A})$, and $Q_A$ be a bounded real-valued function that satisfies the relation*

$$
Q_A(s,a) = \begin{cases} (\mathcal{T}^* Q_A)(s,a) & \text{if } a \in A_s, \\ \tilde{Q}(s,a) & \text{otherwise,} \end{cases} \tag{10}
$$

*for all $s \in \mathcal{S}$ and $a \in \mathcal{A}$. Suppose there is a policy $\pi$ such that for all $s \in \mathcal{S}$,*

$$
V^\pi(s) \geq \sup_{a \in \mathcal{A}} \tilde{Q}(s,a)
$$

*and*

$$Q^\pi(s, a_s) \geq V^\pi(s)$$

*for some $a_s \in A_s$. If there exists a deterministic policy $\pi_A^* \colon \mathcal{S} \to \mathcal{A}$ that is optimal under the constraint $\pi_A(s) \in A_s$ for all $s \in \mathcal{S}$, then*

$$\pi_A^*(s) = \arg\max_{a \in \mathcal{A}} Q_A(s, a).$$

*Proof.* Define $\mathcal{T}_A$ as in Theorem 8. We can see that there is a unique bounded real-valued function $Q_A$ that satisfies (10), because by Theorem 8, $\mathcal{T}_A$ has unique fixed point $Q_A$.

We proceed to prove that for each $s \in \mathcal{S}$, $Q_A(s, \pi_A^*(s)) \geq V^{\pi_A^*}(s)$. Define a sequence $(Q_n)$ of bounded real-valued functions on $\mathcal{S} \times \mathcal{A}$ by the recurrence relation

$$Q_n = \begin{cases} Q_0 & \text{if } n = 0, \\ \mathcal{T}_A Q_{n-1} & \text{otherwise}, \end{cases} \tag{11}$$

where $Q_0 \colon \mathcal{S} \times \mathcal{A} \to \mathbb{R}$ is defined as

$$Q_0(s, a) = \begin{cases} Q^{\pi_A^*}(s, a) & \text{if } a \in A_s, \\ \tilde{Q}(s, a) & \text{otherwise}. \end{cases}$$

When $n = 0$, for all $s \in \mathcal{S}$

$$Q_0(s, \pi_A^*(s)) = Q^{\pi_A^*}(s, \pi_A^*(s)) = V^{\pi_A^*}(s),$$

because $\pi_A^*(s) \in A_s$. Assume $Q_{n-1}(s, \pi_A^*(s)) \geq V^{\pi_A^*}(s)$ for all $s \in \mathcal{S}$. Then for all $s \in \mathcal{S}$,

$$
\begin{aligned}
Q_n(s, \pi_A^*(s)) &= \mathbb{E}_{s' \sim P(s, \pi_A^*(s)), r \sim R(s, \pi_A^*(s), s')} \left[ r + \gamma \sup_{a' \in \mathcal{A}} Q_{n-1}(s', a') \right] \\
&\geq \mathbb{E}_{s' \sim P(s, \pi_A^*(s)), r \sim R(s, \pi_A^*(s), s')} \left[ r + \gamma Q_{n-1}(s', \pi_A^*(s')) \right] \\
&\geq \mathbb{E}_{s' \sim P(s, \pi_A^*(s)), r \sim R(s, \pi_A^*(s), s')} \left[ r + \gamma V^{\pi_A^*}(s') \right] \\
&= \mathbb{E}_{s' \sim P(s, \pi_A^*(s)), r \sim R(s, \pi_A^*(s), s')} \left[ r + \gamma \mathbb{E}_{a' \sim \pi_A^*(s')} \left[ Q^{\pi_A^*}(s', a') \right] \right] \\
&= (\mathcal{T}^{\pi_A^*} Q^{\pi_A^*})(s, \pi_A^*(s)) \\
&= Q^{\pi_A^*}(s, \pi_A^*(s)) \\
&= V^{\pi_A^*}(s).
\end{aligned}
$$

So by mathematical induction, $Q_n(s, \pi_A^*(s)) \geq V^{\pi_A^*}(s)$ for all $s \in \mathcal{S}$ and $n \geq 0$. Therefore,

$$Q_A(s, \pi_A^*(s)) = \lim_{n \to \infty} Q_n(s, \pi_A^*(s)) \geq V^{\pi_A^*}(s).$$

Since for all $s \in \mathcal{S}$ and $a \in A_s$,

$$Q_A(s, a) = (\mathcal{T}_A Q_A)(s, a) = \tilde{Q}(s, a) \leq V^\pi(s).$$

We can define a deterministic policy $\pi_A \colon \mathcal{S} \to \mathcal{A}$ that maps $s \in \mathcal{S}$ to $a_s$. Since $\pi_A(s) = a_s \in A_s$ for all $s \in \mathcal{S}$ and $\pi_A^*$ is the optimal policy among the policies that satisfy this constraint, we have $V^{\pi_A^*} \succeq V^{\pi_A}$. So we may conclude that for all $s \in \mathcal{S}$,

$$\sup_{a \in \mathcal{A} \setminus A_s} Q_A(s, a) \leq V^\pi(s) \leq V^{\pi_A}(s) \leq V^{\pi_A^*}(s) = Q_A(s, \pi_A^*(s)). \tag{12}$$

We finish the proof by showing that for all $s \in \mathcal{S}$,

$$Q_A(s, \pi_A^*(s)) = \max_{a \in \mathcal{A}} Q_A(s, a).$$

Recall the sequence $(Q_n)$ we previously defined by the recurrence relation (11). We will prove that for every $n$, $s \in \mathcal{S}$, and $a \in \mathcal{A}$,

$$Q_n(s, a) \leq V^{\pi_A^*}(s) = Q_n(s, \pi_A^*(s)).$$

Assume $n = 0$. Fix $s^\dagger \in \mathcal{S}$ and $a^\dagger \in \mathcal{A}$. If $a^\dagger \notin A_{s^\dagger}$, then by the observation we made in (12),

$$Q_0(s^\dagger, a^\dagger) = \tilde{Q}(s^\dagger, a^\dagger) \leq V^{\pi_A^*}(s^\dagger).$$

If $a^\dagger \in A_{s^\dagger}$, consider a policy $\pi^\dagger \colon \mathcal{S} \to \mathcal{A}$ defined as

$$\pi^\dagger(s) = \begin{cases} a^\dagger & \text{if } s = s^\dagger, \\ \pi_A^*(s) & \text{otherwise.} \end{cases}$$

For all $s \in \mathcal{S}$, $\pi^\dagger(s) \in A_s$, so $V^{\pi_A^*} \succeq V^{\pi^\dagger}$. If $Q_0(s^\dagger, a^\dagger) < V^{\pi_A^*}(s^\dagger)$, it satisfies our hypothesis. Otherwise,

$$Q^{\pi_A^*}(s^\dagger, \pi^\dagger(s^\dagger)) = Q_0(s^\dagger, a^\dagger) \geq V^{\pi_A^*}(s^\dagger) = Q^{\pi_A^*}(s^\dagger, \pi_A^*(s^\dagger))$$

and for $s \neq s^\dagger$,

$$Q^{\pi_A^*}(s, \pi^\dagger(s)) = Q^{\pi_A^*}(s, \pi_A^*(s)),$$

so by Lemma 9, $V^{\pi^\dagger} \succeq V^{\pi_A^*}$, which means $V^{\pi^\dagger} \equiv V^{\pi_A^*}$. Then

$$
\begin{aligned}
Q^{\pi_A^*}(s^\dagger, a^\dagger) &= (\mathcal{T}^{\pi_A^*} Q^{\pi_\beta^*})(s^\dagger, a^\dagger) \\
&= \mathbb{E}_{s' \sim P(s^\dagger, a^\dagger), r \sim R(s^\dagger, a^\dagger, s')} \left[ r + \gamma Q^{\pi_A^*}(s', \pi_A^*(s')) \right] \\
&= \mathbb{E}_{s' \sim P(s^\dagger, a^\dagger), r \sim R(s^\dagger, a^\dagger, s')} \left[ r + \gamma V^{\pi_A^*}(s') \right] \\
&= \mathbb{E}_{s' \sim P(s^\dagger, a^\dagger), r \sim R(s^\dagger, a^\dagger, s')} \left[ r + \gamma V^{\pi^\dagger}(s') \right] \\
&= \mathbb{E}_{s' \sim P(s^\dagger, a^\dagger), r \sim R(s^\dagger, a^\dagger, s')} \left[ r + \gamma Q^{\pi^\dagger}(s', \pi^\dagger(s')) \right] \\
&= (\mathcal{T}^{\pi^\dagger} Q^{\pi^\dagger})(s^\dagger, a^\dagger) \\
&= Q^{\pi^\dagger}(s^\dagger, \pi^\dagger(s^\dagger)) \\
&= V^{\pi^\dagger}(s^\dagger) \\
&= V^{\pi_A^*}(s^\dagger).
\end{aligned}
$$

So $Q_0(s^\dagger, a^\dagger) \leq V^{\pi_A^*}(s^\dagger)$ in both cases. Since it is obvious that

$$Q_0(s^\dagger, \pi_A^*(s^\dagger)) = Q^{\pi_A^*}(s^\dagger, \pi_A^*(s^\dagger)) = V^{\pi_A^*}(s^\dagger),$$

our hypothesis holds for $n = 0$.

Assume the hypothesis holds for $n - 1$. Fix $s^\dagger \in \mathcal{S}$ and $a^\dagger \in \mathcal{A}$. If $a^\dagger \notin A_{s^\dagger}$,

$$Q_n(s^\dagger, a^\dagger) = (\mathcal{T}_\beta^* Q_{n-1})(s^\dagger, a^\dagger) = Q_\beta^{\text{LB}}(s^\dagger, a^\dagger) \leq V^{\pi_\beta^*}(s^\dagger)$$

by (12). Otherwise,

$$
\begin{aligned}
Q_n(s^\dagger, a^\dagger) &= (\mathcal{T}_A Q_{n-1})(s^\dagger, a^\dagger) \\
&= (\mathcal{T}^* Q_{n-1})(s^\dagger, a^\dagger) \\
&= \mathbb{E}_{s' \sim P(s^\dagger, a^\dagger), r \sim R(s^\dagger, a^\dagger, s')} \left[ r + \gamma \sup_{a' \in \mathcal{A}} Q_{n-1}(s', a') \right] \\
&= \mathbb{E}_{s' \sim P(s^\dagger, a^\dagger), r \sim R(s^\dagger, s^\dagger, s')} \left[ r + \gamma \max_{a' \in \mathcal{A}} Q_{n-1}(s', a') \right] \\
&= \mathbb{E}_{s' \sim P(s^\dagger, a^\dagger), r \sim R(s^\dagger, s^\dagger, s')} \left[ r + \gamma V^{\pi_A^*}(s') \right] \\
&= \mathbb{E}_{s' \sim P(s^\dagger, a^\dagger), r \sim R(s^\dagger, s^\dagger, s')} \left[ r + \gamma \mathbb{E}_{a' \sim \pi_A^*(s')} \left[ Q^{\pi_A^*}(s', a') \right] \right] \\
&= (\mathcal{T}^{\pi_A^*} Q^{\pi_A^*})(s^\dagger, a^\dagger) \\
&= Q^{\pi_A^*}(s^\dagger, a^\dagger) \\
&= Q_0(s^\dagger, a^\dagger) \\
&\leq V^{\pi_A^*}(s^\dagger).
\end{aligned}
$$

When $a^\dagger = \pi_A^*(s^\dagger)$, the inequality becomes equality. So by mathematical induction, for every $n$, $s \in \mathcal{S}$, and $a \in \mathcal{A}$,

$$Q_n(s,a) \leq V^{\pi_A^*}(s) = Q_n(s, \pi_A^*(s)).$$

Sending $n$ to infinity, we can see that for all $s \in \mathcal{S}$ and $a \in \mathcal{A}$,

$$Q_A(s,a) = \lim_{n\to\infty} Q_n(s,a) \leq V^{\pi_A^*}(s) = \lim_{n\to\infty} Q_n(s, \pi_A^*(s)) = Q_A(s, \pi_A^*(s)).$$

Therefore,

$$Q_A(s, \pi_A^*(s)) = \max_{a \in \mathcal{A}} Q_A(s,a).$$

$\square$

**Theorem 11.** *Any initial bounded real-valued function on $\mathcal{S} \times \mathcal{A}$ can converge to a unique fixed point $Q_\beta^*$ by repeatedly applying $\mathcal{T}_\beta^*$. Suppose for each $s \in \mathcal{S}$,*

$$Q^\beta(s, a_s) \geq \mathbb{E}_{a \sim \beta(s)}\left[Q^\beta(s,a)\right]$$

*for some $a_s \in \mathcal{A} \setminus \mathrm{OOD}(s)$. If there exists a deterministic policy $\pi_\beta^*\colon \mathcal{S} \to \mathcal{A}$ that is optimal under the constraint $\pi(s) \notin \mathrm{OOD}(s)$ for all $s \in \mathcal{S}$, then $\pi_\beta^*(s) = \arg\max_{a \in \mathcal{A}} Q_\beta^*(s,a)$ for all $s \in \mathcal{S}$.*

*Proof.* First observe that for all $s \in \mathcal{S}$ and $a \in \mathcal{A}$,

$$Q_\beta^{\mathrm{LB}}(s,a) \geq \frac{r_{\min}}{1 - \gamma},$$

and

$$Q_\beta^{\mathrm{LB}}(s,a) \leq V^\beta(s) - r_{\max} + r_{\min} \leq \frac{r_{\max}}{1 - \gamma} - r_{\max} + r_{\min},$$

by Lemma 6. This implies $Q_\beta^{\mathrm{LB}} \in L^\infty(\mathcal{S} \times \mathcal{A})$. Since $\tilde{Q} = Q_\beta^{\mathrm{LB}}$ and $A_s = \mathcal{A} \setminus \mathrm{OOD}(s)$ satisfies the conditions of Lemma 8, any initially bounded real-valued function on $\mathcal{S} \times \mathcal{A}$ converges to a unique fixed point, which we denote by $Q_\beta^*$, through repeated application of $\mathcal{T}_A$, which is in fact, $\mathcal{T}_\beta^*$.

For all $s \in \mathcal{S}$,

$$Q_\beta^{\mathrm{LB}}(s,a) \leq V^\beta(s) - r_{\max} + r_{\min} \leq V^\beta(s),$$

which means

$$\sup_{a \in \mathcal{A}} Q_\beta^{\mathrm{LB}}(s,a). \leq V^\beta(s) = \mathbb{E}_{a \sim \beta(s)}\left[Q^\beta(s,a)\right] < \sup_{a \in \mathcal{A} \setminus \mathrm{OOD}(s)} Q^\beta(s,a).$$

Now we can see that the second part of the theorem is a special case of Theorem 10, where $A_s = \mathcal{A} \setminus \mathrm{OOD}(s)$, $\tilde{Q} = Q_\beta^{\mathrm{LB}}$, $Q_A = Q_\beta^*$, $\pi = \beta$, and $\pi_A^* = \pi_\beta^*$. $\square$

**Lemma 12.** *Let $\Pi = \{\pi_0, \pi_1, \pi_2, \dots, \pi_{N-1}\}$ be a finite set of policies. If $\pi^*$ is a policy such that for each $s \in \mathcal{S}$, there is $i \in [N]$ such that $\pi^*(s) = \pi_i(s)$ and $V^{\pi_i}(s) = \max_{\pi \in \Pi} V^\pi(s)$, then $V^{\pi^*} \succeq V^\pi$ for every $\pi \in \Pi$.*

*Proof.* Define a sequence $(Q_n)$ of bounded real-valued functions by the recurrence relation

$$Q_n = \begin{cases} \max_{\pi \in \Pi} Q^\pi & \text{if } n = 0, \\ \mathcal{T}^{\pi^*} Q_{n-1}, & \text{otherwise.} \end{cases}$$

We want to show that $Q_n \succeq \max_{\pi \in \Pi} Q^\pi$ for all $n \geq 0$. The base case is trivial. Assume $Q_{n-1} \succeq \max_{\pi \in \Pi} Q^\pi$. For each $s \in \mathcal{S}$, there is $i \in [N]$ such that $\pi^*(s) = \pi_i(s)$ and $V^{\pi_i}(s) = \max_{\pi \in \Pi} V^\pi(s)$, which implies

$$\mathbb{E}_{a \sim \pi^*(s)}\left[Q_{n-1}(s,a)\right] \geq \mathbb{E}_{a \sim \pi_i(s)}\left[Q^{\pi_i}(s)\right] = V^{\pi_i}(s) = \max_{\pi \in \Pi} V^\pi(s).$$

Now for all $s \in \mathcal{S}$ and $a \in \mathcal{A}$,

$$
\begin{aligned}
Q_n(s,a) &= (\mathcal{T}^{\pi^*} Q_{n-1})(s,a) \\
&= \mathbb{E}_{s' \sim P(s,a), r \sim R(s,a,s')} \left[ r + \gamma \, \mathbb{E}_{a' \sim \pi^*(s')} \left[ Q_{n-1}(s',a') \right] \right] \\
&\geq \mathbb{E}_{s' \sim P(s,a), r \sim R(s,a,s')} \left[ r + \gamma \max_{\pi \in \Pi} V^\pi(s') \right] \\
&= \max_{\pi \in \Pi} \mathbb{E}_{s' \sim P(s,a), r \sim R(s,a,s')} \left[ r + \gamma \, \mathbb{E}_{a' \sim \pi(s')} \left[ Q^\pi(s',a') \right] \right] \\
&= \max_{\pi \in \Pi} (\mathcal{T}^\pi Q^\pi)(s,a) \\
&= \max_{\pi \in \Pi} Q^\pi(s,a).
\end{aligned}
$$

By mathematical induction, $Q_n \succeq \max_{\pi \in \Pi} Q^\pi$ for all $n \geq 0$. Therefore,

$$
Q^{\pi^*}(s,a) = \lim_{n \to \infty} Q_n(s,a) \geq \max_{\pi \in \Pi} Q^\pi(s,a)
$$

for all $s \in \mathcal{S}$ and $a \in \mathcal{A}$.

Fix $s \in \mathcal{S}$. There is $i \in [N]$ such that $\pi^*(s) = \pi_i(s)$ and $V^{\pi_i}(s) = \max_{\pi \in \Pi} V^\pi(s)$. Then

$$
V^{\pi^*}(s) = \mathbb{E}_{a \sim \pi^*(s)} \left[ Q^{\pi^*}(s,a) \right] \geq \mathbb{E}_{a \sim \pi_i(s)} \left[ Q^{\pi_i}(s,a) \right] = V^{\pi_i}(s) = \max_{\pi \in \Pi} V^\pi(s).
$$

Our choice of $s$ was arbitrary, so $V^{\pi^*} \succeq \max_{\pi \in \Pi} V^\pi$. $\qquad\square$

**Theorem 13.** *Any initial bounded real-valued function on $\mathcal{S} \times \mathcal{A}$ can converge to a unique fixed point $Q_\mathcal{B}^*$ by repeatedly applying $\mathcal{T}_\mathcal{B}^*$. Suppose for each $\beta \in \mathcal{B}$ and $s \in \mathcal{S}$,*

$$
Q^\beta(s, a_s^\beta) \geq \mathbb{E}_{a \sim \beta(s)} \left[ Q^\beta(s,a) \right]
$$

*for some $a_s^\beta \in \mathcal{A} \setminus \mathrm{OOD}(s)$. If there exists a deterministic policy $\pi_\mathcal{B}^* \colon \mathcal{S} \to \mathcal{A}$ that is optimal under the constraint $\pi(s) \notin \mathrm{OOD}(s)$ for all $s \in \mathcal{S}$, then $\pi_\mathcal{B}^*(s) = \arg\max_{a \in \mathcal{A}} Q_\mathcal{B}^*(s,a)$ for all $s \in \mathcal{S}$.*

*Proof.* First observe that for all $s \in \mathcal{S}$, $a \in \mathcal{A}$, and $\beta \in \mathcal{B}$,

$$
Q_\beta^{\mathrm{LB}}(s,a) \geq \frac{r_{\min}}{1 - \gamma},
$$

and

$$
Q_\beta^{\mathrm{LB}}(s,a) \leq V^\beta(s) - r_{\max} + r_{\min} \leq \frac{r_{\max}}{1 - \gamma} - r_{\max} + r_{\min},
$$

by Lemma 6. So obviously,

$$
Q_\mathcal{B}^{\mathrm{LB}}(s,a) = \max_{\beta \in \mathcal{B}} Q_\beta^{\mathrm{LB}}(s,a) \geq \frac{r_{\min}}{1 - \gamma},
$$

and

$$
Q_\mathcal{B}^{\mathrm{LB}}(s,a) = \max_{\beta \in \mathcal{B}} Q_\beta^{\mathrm{LB}}(s,a) \leq \frac{r_{\max}}{1 - \gamma} - r_{\max} + r_{\min},
$$

for all $s \in \mathcal{S}$ and $\mathcal{A}$. This implies $Q_\mathcal{B}^{\mathrm{LB}} \in L^\infty(\mathcal{S} \times \mathcal{A})$. Since $\tilde{Q} = Q_\mathcal{B}^{\mathrm{LB}}$ and $A_s = \mathcal{A} \setminus \mathrm{OOD}(s)$ satisfies the conditions of Lemma 8, any initially bounded real-valued function on $\mathcal{S} \times \mathcal{A}$ converges to a unique fixed point, which we denote by $Q_\mathcal{B}^*$, through repeated application of $\mathcal{T}_A$, which is in fact, $\mathcal{T}_\mathcal{B}^*$.

For each $\beta \in \mathcal{B}$ define a deterministic policy $\beta' \colon \mathcal{S} \to \mathcal{A}$ so that $\beta'(s) = a_s^\beta$ for each $s \in \mathcal{S}$. Then

$$
Q^\beta(s, \beta'(s)) = Q^\beta(s, a_s^\beta) \geq \mathbb{E}_{a \sim \beta(s)} \left[ Q^\beta(s,a) \right]
$$

for all $s \in \mathcal{S}$, so $V^{\beta'} \succeq V^\beta$ by Lemma 9. We will denote the set $\{ \beta' : \beta \in \mathcal{B} \}$ by $\mathcal{B}'$. Consider a policy $\beta^* \colon \mathcal{S} \to \mathcal{A}$ defined as

$$
\beta^*(s) = \left( \arg\max_{\beta' \in \mathcal{B}'} V^{\beta'}(s) \right)(s),
$$

that is, for each state, we follow the $\beta'$ with the highest value. Obviously, $\beta^*(s) \in \mathcal{A} \setminus \text{OOD}(s)$ for all $s \in \mathcal{S}$, and by Lemma 12, for all $s \in \mathcal{S}$, $a \in \mathcal{A}$, and $\beta \in \mathcal{B}$,

$$Q_\beta^{\text{LB}}(s, a) \le V^\beta(s) - r_{\max} + r_{\min} \le V^\beta(s) \le V^{\beta'}(s) \le V^{\beta^*}(s),$$

which means

$$V^{\beta^*}(s) \ge \sup_{a \in \mathcal{A}} \max_{\beta \in \mathcal{B}} Q_\beta^{\text{LB}}(s, a) = \sup_{a \in \mathcal{A}} Q_{\mathcal{B}}^{\text{LB}}(s, a).$$

Now we can see that the second part of the theorem is a special case of Theorem 10, where $A_s = \mathcal{A} \setminus \text{OOD}(s)$, $\tilde{Q} = Q_{\mathcal{B}}^{\text{LB}}$, $Q_A = Q_{\mathcal{B}}^*$, $\pi = \beta^*$, and $\pi_A^* = \pi_\beta^*$. $\qquad\square$

**Proposition 14.** *Let $\mathbf{X}$ be a non-degenerate multivariate Gaussian random vector with mean $\boldsymbol{\mu} \in \mathbb{R}^d$ and a diagonal covariance matrix $\text{diag}(\boldsymbol{\sigma})^2 \in \mathbb{R}^{d \times d}$. For $\mathbf{y} \in \mathbb{R}^d$,*

$$\mathbb{E}\left[\|\mathbf{X} - \mathbf{y}\|_1\right] = \sum_{i=1}^d \left[(y_i - \mu_i)\,\text{erf}\left(\frac{y_i - \mu_i}{\sigma_i\sqrt{2}}\right) + \sqrt{\frac{2}{\pi}}\sigma_i \exp\left(-\frac{(y_i - \mu_i)^2}{2\sigma_i^2}\right).\right]$$

*Proof.* Let $\mathbf{X} = (X_1, X_2, \ldots, X_d)$, $\mathbf{y} = (y_1, y_2, \ldots, y_d)$, $\boldsymbol{\mu} = (\mu_1, \mu_2, \ldots, \mu_d)$, and $\boldsymbol{\sigma} = (\sigma_1, \sigma_2, \ldots, \sigma_d)$. We may assume that $\sigma_1, \sigma_2, \ldots, \sigma_d > 0$. Then

$$\mathbb{E}\left[\|\mathbf{X} - \mathbf{y}\|_1\right] = \mathbb{E}\left[\sum_{i=1}^d |X_i - y_i|\right] = \sum_{i=1}^d \mathbb{E}\left[|X_i - y_i|\right].$$

Define $g_i(y) = \mathbb{E}\left[|X_i - y|\right]$.

$$g_i'(y) = \mathbb{E}\left[\frac{\text{d}}{\text{d}y}|X_i - y|\right] = \mathbb{E}[\mathbf{1}_{X_i < y} - \mathbf{1}_{X_i > y}] = F_{X_i}(y) - (1 - F_{X_i}(y)) = 2F_{X_i}(y) - 1,$$

where $F_{X_i}$ is the cumulative distribution function of $X_i$. So

$$g_i'(y) = \text{erf}\left(\frac{y - \mu_i}{\sigma_i\sqrt{2}}\right).$$

Observe that

$$\begin{aligned}
g_i(\mu_i) &= \mathbb{E}\left[|X_i - \mu_i|\right]\\
&= \frac{1}{\sqrt{2\pi}\sigma_i}\int_{\mu_i}^\infty (x_i - \mu_i)\exp\left(-\frac{(x_i - \mu_i)^2}{2\sigma_i^2}\right)\,\text{d}x_i\\
&\quad - \frac{1}{\sqrt{2\pi}\sigma_i}\int_{-\infty}^{\mu_i} (x_i - \mu_i)\exp\left(-\frac{(x_i - \mu_i)^2}{2\sigma_i^2}\right)\,\text{d}x_i.
\end{aligned}$$

Substituting $u_i = (x_i - \mu_i)/\sigma_i$,

$$\begin{aligned}
g_i(\mu_i) &= \frac{1}{\sqrt{2\pi}\sigma_i}\left[\int_0^\infty \sigma_i u_i e^{-\frac{1}{2}u_i^2}\sigma_i\,\text{d}u_i - \int_{-\infty}^0 \sigma_i u_i e^{-\frac{1}{2}u_i^2}\sigma_i\,\text{d}u_i\right]\\
&= \frac{\sigma_i}{\sqrt{2\pi}}\left[\int_0^\infty u_i e^{-\frac{1}{2}u_i^2}\,\text{d}u_i - \int_{-\infty}^0 u_i e^{-\frac{1}{2}u_i^2}\,\text{d}u_i\right]\\
&= \sqrt{\frac{2}{\pi}}\sigma_i.
\end{aligned}$$

By the fundamental theorem of calculus,

$$g_i(y) = g_i(\mu_i) + \int_{\mu_i}^y g_i'(v)\,\text{d}v = \sqrt{\frac{2}{\pi}}\sigma_i + \int_{\mu_i}^y \text{erf}\left(\frac{v - \mu_i}{\sigma_i\sqrt{2}}\right)\,\text{d}v.$$

Substituting $z = (v - \mu_i)/(\sigma_i\sqrt{2})$,

$$\begin{aligned}
\int_{\mu_i}^y \text{erf}\left(\frac{v - \mu_i}{\sigma_i\sqrt{2}}\right)\,\text{d}v &= \int_0^{(y-\mu_i)/(\sigma_i\sqrt{2})} \text{erf}(z)\sigma_i\sqrt{2}\,\text{d}z\\
&= \sqrt{2}\sigma_i\left[\left(\frac{y - \mu_i}{\sigma_i\sqrt{2}}\right)\text{erf}\left(\frac{y - \mu_i}{\sigma_i\sqrt{2}}\right) + \frac{1}{\sqrt{\pi}}\exp\left(-\frac{(y - \mu_i)^2}{2\sigma_i^2}\right) - \frac{1}{\sqrt{\pi}}\right].
\end{aligned}$$

Therefore,

$$g_i(y) = (y - \mu_i) \operatorname{erf}\left(\frac{y - \mu_i}{\sigma_i\sqrt{2}}\right) + \sqrt{\frac{2}{\pi}}\sigma_i \exp\left(-\frac{(y - \mu_i)^2}{2\sigma_i^2}\right),$$

which implies

$$\mathbb{E}\left[\|\mathbf{X} - \mathbf{y}\|_1\right] = \sum_{i=1}^d g_i(y_i) = \sum_{i=1}^d \left[(y_i - \mu_i) \operatorname{erf}\left(\frac{y_i - \mu_i}{\sigma_i\sqrt{2}}\right) + \sqrt{\frac{2}{\pi}}\sigma_i \exp\left(-\frac{(y_i - \mu_i)^2}{2\sigma_i^2}\right).\right]$$

□

## C  PRACTICAL ALGORITHM

### C.1  STAGE I: BEHAVIOUR POLICY LEARNING

In theory, each behaviour policy is well-defined on every state $s \in \mathcal{S}$. However, in practice, we can trust our estimations only in the vicinity of the states they were trained on. The problem is, we train each $\hat{\beta}$ only on the states they are assigned to. Therefore, we need a mechanism to determine which behaviour policy estimates we can trust given a state $s \in \mathcal{S}$. For this purpose, we additionally train a classifier $f_\psi \colon \mathcal{S} \to \mathcal{P}_d([K])$ using the computed assignments and determine the credible set by the equation

$$\hat{\mathcal{V}}(s) = \left\{\hat{\beta}_i \in \hat{\mathcal{B}} : f_\psi(i \mid s) \geq \frac{1}{b}\max_{j \in [K]} f_\psi(j \mid s)\right\},$$

where $b > 0$ is a hyperparameter. We accordingly modify the definition of $Q_{\hat{\mathcal{B}}}^{\mathrm{LB}}$ to

$$Q_{\hat{\mathcal{B}}}^{\mathrm{LB}}(s, a) = \max_{\hat{\beta} \in \hat{\mathcal{V}}(s)} Q_{\hat{\beta}}^{\mathrm{LB}}(s, a),$$

for all $s \in \mathcal{S}$ and $a \in \mathcal{A}$.

### C.2  STAGE II: BEHAVIOUR VALUE LEARNING

To learn the value functions of all $K$ behaviour policies in parallel, we leverage a network $V_\zeta \colon \mathcal{S} \to \mathbb{R}^K$ with $K$ outputs. The per sample temporal difference (TD) loss function can be written by the equation

$$\ell_V(\zeta \,; s, r, s') = \left(V_\zeta(s)[\mathbb{A}(s)] - r - \gamma V_{\zeta'}(s')[\mathbb{A}(s')]\right)^2,$$

where $s$, $r$, and $s'$ are the state, reward, and next state sampled from the dataset, respectively, $\zeta'$ is the target network parameter that is updated by polyak averaging as in Lillicrap et al. (2016), and $V_\zeta(s)[i]$ is the $i$-th coordinate of $V_\zeta(s)$. Note that $\mathbb{A}(s)$, the cluster assignment of $s$, is equal to $\mathbb{A}(s')$, because we assign each trajectory to the same cluster.

### C.3  STAGE III: POLICY LEARNING

In practice, it is infeasible to compute the supremum term in the penalised Bellman operator $\mathcal{T}_{\mathcal{B}}^*$. We instead adopt the actor-critic formulation that alternates between the policy improvement step and the critic learning step. The goal of the policy improvement step is to find an action that maximises the critic for each state. The challenge is that the critic is highly non-convex due to the penalisation of critic values for actions between the means of the behaviour policies, causing gradient methods to yield suboptimal solutions. Hence, we search for the optimal action in the vicinity of each behaviour policy's mean simultaneously. This is done by training a network to output not the optimal action itself but the difference between the optimal action and one of the behaviour policy means. Utilising an ensemble of $K$ neural networks $g_{\psi_\pi^0}, g_{\psi_\pi^1}, \dots, g_{\psi_\pi^{K-1}}$, we can compute $K$ action candidates $a_0, a_1, \dots, a_{K-1}$, where

$$a_i = \boldsymbol{\mu}_{\hat{\beta}_i}(s) + g_{\psi_\pi^i}(s),$$

and $\boldsymbol{\mu}_{\hat{\beta}}$ is the mean vector of the behaviour policy estimate $\hat{\beta}$. Then we choose the best action $a_{i_s^*}$ with respect to the current critic function $Q_{\psi_Q}$, that is,

$$i_s^* = \underset{i \in \hat{\mathcal{I}}(s)}{\arg\max}\, Q_{\psi_Q}(s, a_i),$$

where

$$\hat{\mathcal{I}}(s) = \left\{ i \in [K] : \hat{\beta}_i \in \hat{\mathcal{V}}(s) \right\}.$$

The loss function for the policy improvement step can be written by the following equation:

$$\ell_\pi(\psi_\pi \, ; s) = - \max_{i \in \hat{\mathcal{I}}^*} Q_{\psi_Q} \left( s, \boldsymbol{\mu}_{\hat{\beta}_i}(s) + g_{\psi_\pi^i}(s) \right).$$

The critic learning step has two objectives: minimising the TD error and penalising the OOD actions. For the first objective, we adopt the conventional TD loss adapted to match the way we defined our policy, which is represented by the equation

$$\ell_Q^{\mathrm{TD}}(\psi_Q \, ; s, a, r, s') = \left( Q_{\psi_Q}(s, a) - T(r, s') \right)^2,$$

where $s$, $a$, $r$, and $s'$ are the state, action, reward, and next state sampled from the dataset, respectively, and the TD target $T(r, s')$ is defined as

$$T(r, s') = r + \gamma \max_{\hat{\beta} \in \hat{\mathcal{V}}(s)} Q_{\psi_Q'} \left( s', \boldsymbol{\mu}_{\hat{\beta}}(s') + g_{\psi_\pi'}(s') \right).$$

$\psi_Q'$ and $\psi_\pi'$ in the preceding equation are the target critic network parameters and target actor network parameters, respectively, which are updated by polyak averaging to gradually follow $\psi_Q$ and $\psi_\pi$.

The second objective of the critic learning step is to penalise the critic values of OOD actions towards $Q_{\hat{\mathcal{B}}}^{\mathrm{LB}}(s, a)$. In order to compute $Q_{\hat{\mathcal{B}}}^{\mathrm{LB}}$, we need to compute $Q_{\hat{\beta}}^{\mathrm{LB}}$ for each $\hat{\beta} \in \hat{\mathcal{B}}$. However, computing $Q_{\hat{\beta}}^{\mathrm{LB}}$ is not straightforward, due to the term $\mathbb{E}_{a' \sim \hat{\beta}(s)}[\|a - a'\|]$ in (7). We discovered that if we use a 1-norm and assume that $\hat{\beta}$ has a diagonal covariance matrix, the expectation has the following closed-form expression (Proposition 14):

$$\mathbb{E}_{a' \sim \hat{\beta}(s)} \left[ \|a - a'\|_1 \right] = \sum_{i=1}^{d_a} (y_i - \mu_i(s)) \operatorname{erf} \left( \frac{a_i - \mu_i(s)}{\sigma_i(s)\sqrt{2}} \right) + \sqrt{\frac{2}{\pi}} \sum_{i=1}^{d_a} \sigma_i \exp \left( - \frac{(a_i - \mu_i(s))^2}{2\sigma_i^2(s)} \right),$$

where $a = (a_1, a_2, \ldots, a_{d_a})$ and $\hat{\beta}(s)$ is a Gaussian distribution with a state-dependent mean vector $\boldsymbol{\mu}(s) = (\mu_1(s), \mu_2(s), \ldots, \mu_{d_a}(s))$ and a state-dependent covariance matrix whose main diagonal is $\boldsymbol{\sigma}(s) = (\sigma_1(s), \sigma_2(s), \ldots, \sigma_{d_a}(s))$. For $r_{\min}$ and $r_{\max}$, following Mao et al. (2023), we estimate them by the minimum and maximum rewards in all of the datasets of a given task, that is, for example $r_{\min}$ and $r_{\max}$ for a hopper-v2 dataset is computed by the minimum and maximum of the rewards in hopper-expert-v2, hopper-medium-v2, and hopper-random-v2. To sum up, the loss function for regularisation is

$$\ell_Q^{\mathrm{reg}}(\psi_Q \, ; \tilde{s}, \tilde{a}) \quad = \mathbf{1}_{\tilde{a} \in \mathrm{OOD}(\tilde{s})} \left( Q_{\psi_Q}(\tilde{s}, \tilde{a}) - Q_{\hat{\mathcal{B}}}^{\mathrm{LB}}(\tilde{s}, \tilde{a}) \right)^2,$$

where $\tilde{s}$ is a state sampled from the dataset and $\tilde{a}$ is an action sampled from $\pi_{\mathrm{alg}}$. The resulting total loss can be written by the following equation

$$\ell_Q(\psi_Q) = \mathbb{E}_{(s,a,r,s') \sim \mathcal{D}} \left[ \ell_Q^{\mathrm{TD}}(\psi_Q \, ; s, a, r, s') \right] + w_Q \, \mathbb{E}_{\tilde{s} \sim \mathcal{D}, \tilde{a} \sim \pi_{\mathrm{alg}}(\cdot|\tilde{s})} \left[ \ell_Q^{\mathrm{reg}}(\psi_Q \, ; \tilde{s}, \tilde{a}) \right], \quad (13)$$

where $w_Q$ is a tunable hyperparameter. For $\pi_{\mathrm{alg}}$ we adopted

$$\pi_{\mathrm{alg}}(a \mid s) = \frac{1}{2}\tilde{\beta}(a \mid s) + \frac{1}{2}\pi(a \mid s),$$

where $\pi$ is the current policy and $\tilde{\beta}$ is defined as

$$\tilde{\beta}(\cdot \mid s) = \mathcal{N}(\boldsymbol{\mu}_{\hat{\beta}^*}(s), \kappa^2 \boldsymbol{\Sigma}_{\hat{\beta}^*}(s)).$$

Here, $\boldsymbol{\mu}_{\hat{\beta}^*}(s)$ and $\boldsymbol{\Sigma}_{\hat{\beta}^*}(s)$ are the mean vector and covariance matrix of the selected behaviour policy $\hat{\beta}^*$ and $\kappa$ is a tunable hyperparameter, which we set to 2 in most of our experiments. The first term regularises the critic values over a broad range of actions to guide a randomly initialised network $g_{\psi_\pi}$ towards producing near-zero values. The second term regularises the critic values in the vicinity of the current policy allowing delicate control near the boundary of $\mathrm{OOD}(s)$.

### C.4 Implementation Details

We standardised the observations following Fujimoto & Gu (2021) and normalised the rewards so that the average empirical discounted return becomes $\hat{V}$, which we set to 100 for most of our experiments. The algorithm was implemented upon the JAX (Bradbury et al., 2018) framework using the Flax (Heek et al., 2024) library. The scikit-learn (Pedregosa et al., 2011) library was used to compute ARIs and NMIs for Sections 5.3 and D.2.

## D Experiment Details

### D.1 Motivation Experiment

For the experiment in Section 3, we created a single-state infinite-horizon MDP with $\gamma = 0.9$ and a deterministic reward function

$$R(\mathbf{a}) = \frac{1}{2}\exp\left(\|\mathbf{a} - \mathbf{a}^*\|_2^2\right).$$

The optimal action is obviously $\mathbf{a}^*$ and the optimal discounted return is $1/(1-\gamma) = 10$. Since it is a single-state MDP, we adopted a trainable six-dimensional vector instead of a policy network. Similarly, the critic network $Q_\psi \colon \mathcal{A} \to \mathbb{R}$ only takes the action as its input. The loss function functions for the two algorithms are

$$\ell_{\mathrm{IS}}(\psi) = \mathbb{E}_{(r,\mathbf{a})\sim\mathcal{D}}\left[\left(Q_\psi(\mathbf{a}) - r - \gamma Q_{\bar{\psi}}(\mathbf{a}_\phi)\right)^2 + \rho_\phi(\mathbf{a})Q_\psi(\mathbf{a})^2 + \alpha\,\mathbb{E}_{\mathbf{a}'\sim\mathcal{N}(\mathbf{a},\sigma^2 I_6)}\left[Q_\psi(\mathbf{a}')^2\right]\right]$$

$$\ell_{\mathrm{HT}}(\psi) = \mathbb{E}_{(r,\mathbf{a})\sim\mathcal{D}}\left[\left(Q_\psi(\mathbf{a}) - r - \gamma Q_{\bar{\psi}}(\mathbf{a}_\phi)\right)^2 + \alpha\,\mathbb{E}_{\mathbf{a}'\sim\pi_{\mathrm{alg}}}\left[\mathbf{1}_{\mathbf{a}'\in\mathrm{OOD}}Q_\psi(\mathbf{a}')^2\right]\right],$$

where $\mathbf{a}_\phi$ is the current policy, $\bar{\psi}$ is the target critic parameter updated in an EMA fashion, $\rho_\phi(\mathbf{a})$ is the importance sampling ratio defined as

$$\rho_\phi(\mathbf{a}) = \frac{\mu_\phi(\mathbf{a})}{\beta(\mathbf{a})} = \frac{1}{\sigma^6}\exp\left(\frac{1}{2}\|\mathbf{a}\|^2 - \frac{1}{2\sigma^2}\|\mathbf{a} - \mathbf{a}_\phi\|^2\right),$$

$\pi_{\mathrm{alg}}$ is the sampling policy defined as

$$\pi_{\mathrm{alg}} = \frac{1}{2}\mathcal{N}(\mathbf{0}, I_6) + \frac{1}{2}\mathcal{N}(\mathbf{a}_\phi, \sigma^2 I_6),$$

and

$$\mathrm{OOD} = \left\{\mathbf{a} \in \mathbb{R}^6 : \|\mathbf{a}\|_2^2 \leq F_{\chi_6^2}^{-1}(0.5)\right\}.$$

For the actor loss, we used

$$\ell_\pi(\phi) = -Q_\psi(\mathbf{a}_\phi)$$

in both cases.

### D.2 Additional Analysis on the Trajectory Clustering Algorithm

Aside from the three random-medium-expert datasets mentioned in Section 5.3, we also created custom D4RL datasets by concatenating medium and expert datasets of halfcheetah, hopper, and walker2d tasks. The mean and standard deviation of ARIs and NMIs for each configuration over 5 different seeds are reported in Table 7. The configuration $(\lambda_T, \lambda_R) = (1, 0)$ performs the best on average even after we include the three medium-expert datasets.

We also provide visualisations of our clustering results on different datasets of the D4RL benchmark (Figures 4–9). Each row represents a different trajectory in the dataset.

Table 7: The impact of hyperparameters $\lambda_T$ and $\lambda_R$ on the performance of our trajectory clustering algorithm evaluated on custom D4RL datasets. The performance is measured in terms of adjusted rand index (ARI) and normalised mutual information score (NMI).

| | | $\lambda_T = 1$ | | $\lambda_T = 0$ | |
| --- | --- | --- | --- | --- | --- |
| | | $\lambda_R = 1$ | $\lambda_R = 0$ | $\lambda_R = 1$ | $\lambda_R = 0$ |
| halfcheetah-medium-expert | | | | | |
| | ARI | $1.00 \pm 0.00$ | $1.00 \pm 0.00$ | $1.00 \pm 0.00$ | $1.00 \pm 0.00$ |
| | NMI | $1.00 \pm 0.00$ | $0.99 \pm 0.01$ | $1.00 \pm 0.00$ | $1.00 \pm 0.00$ |
| halfcheetah-random-medium-expert | | | | | |
| | ARI | $0.97 \pm 0.04$ | $0.99 \pm 0.00$ | $0.91 \pm 0.19$ | $0.91 \pm 0.17$ |
| | NMI | $0.97 \pm 0.03$ | $0.97 \pm 0.01$ | $0.93 \pm 0.12$ | $0.92 \pm 0.12$ |
| hopper-medium-expert | | | | | |
| | ARI | $0.80 \pm 0.44$ | $1.00 \pm 0.00$ | $0.99 \pm 0.01$ | $1.00 \pm 0.00$ |
| | NMI | $0.80 \pm 0.42$ | $1.00 \pm 0.00$ | $0.97 \pm 0.03$ | $1.00 \pm 0.00$ |
| hopper-random-medium-expert | | | | | |
| | ARI | $0.49 \pm 0.29$ | $0.98 \pm 0.02$ | $0.59 \pm 0.33$ | $0.97 \pm 0.06$ |
| | NMI | $0.57 \pm 0.26$ | $0.97 \pm 0.02$ | $0.66 \pm 0.29$ | $0.98 \pm 0.04$ |
| walker2d-medium-expert | | | | | |
| | ARI | $0.99 \pm 0.01$ | $1.00 \pm 0.00$ | $0.99 \pm 0.02$ | $1.00 \pm 0.00$ |
| | NMI | $0.99 \pm 0.02$ | $1.00 \pm 0.01$ | $0.98 \pm 0.04$ | $1.00 \pm 0.00$ |
| walker2d-random-medium-expert | | | | | |
| | ARI | $0.88 \pm 0.16$ | $0.98 \pm 0.05$ | $0.98 \pm 0.03$ | $1.00 \pm 0.00$ |
| | NMI | $0.90 \pm 0.11$ | $0.98 \pm 0.03$ | $0.97 \pm 0.04$ | $1.00 \pm 0.00$ |
| Average | ARI | $0.86 \pm 0.27$ | $0.99 \pm 0.02$ | $0.91 \pm 0.21$ | $0.98 \pm 0.07$ |
| | NMI | $0.87 \pm 0.24$ | $0.98 \pm 0.02$ | $0.92 \pm 0.17$ | $0.98 \pm 0.06$ |

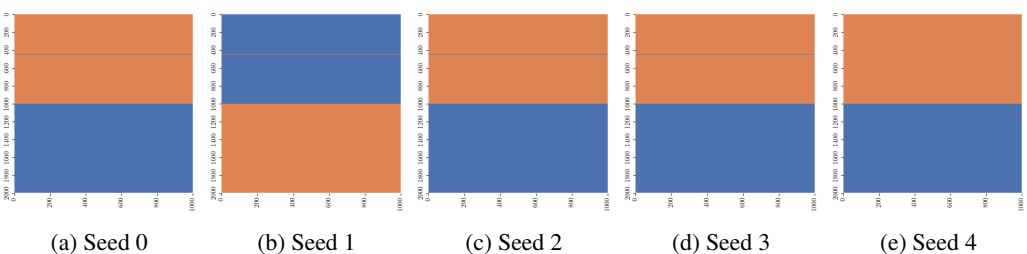

(a) Seed 0  (b) Seed 1  (c) Seed 2  (d) Seed 3  (e) Seed 4

Figure 4: Clustering visualisations for halfcheetah-medium-expert-v2

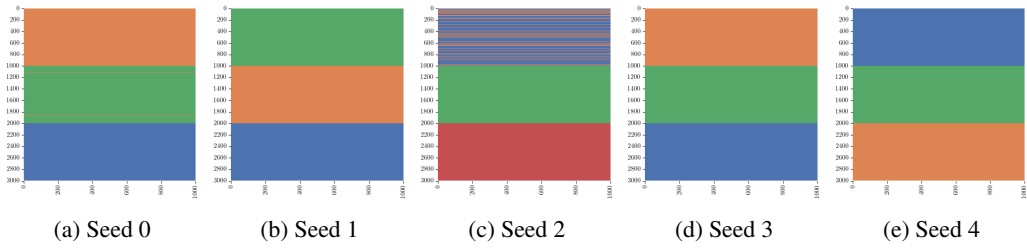

(a) Seed 0  (b) Seed 1  (c) Seed 2  (d) Seed 3  (e) Seed 4

Figure 5: Clustering visualisations for halfcheetah-random-medium-expert-v2

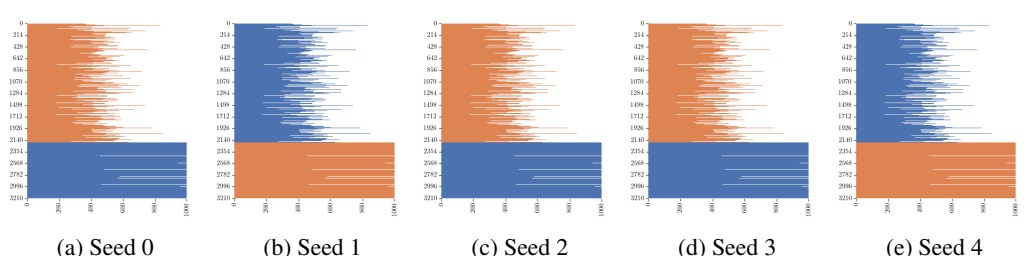

Figure 6: Clustering visualisations for hopper-medium-expert-v2

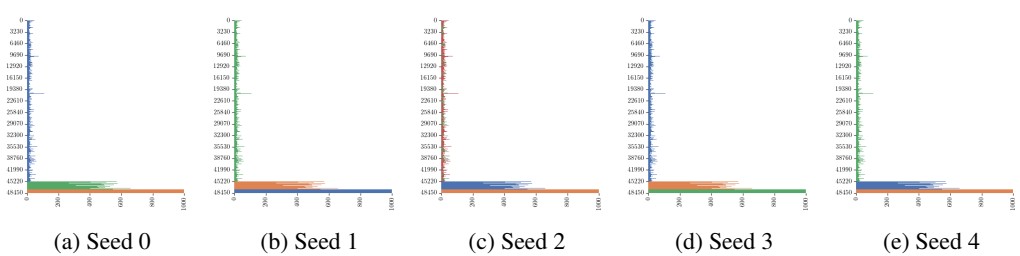

Figure 7: Clustering visualisations for hopper-random-medium-expert-v2

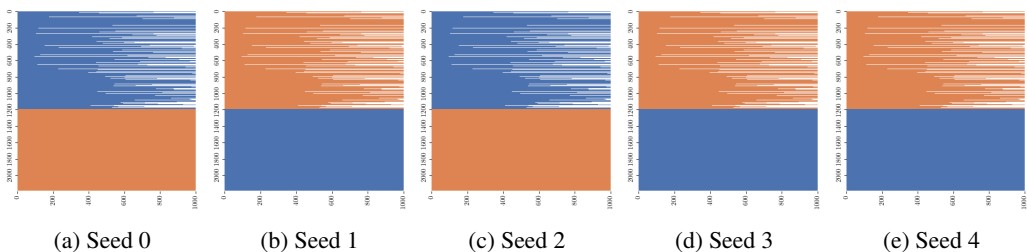

Figure 8: Clustering visualisations for walker2d-medium-expert-v2

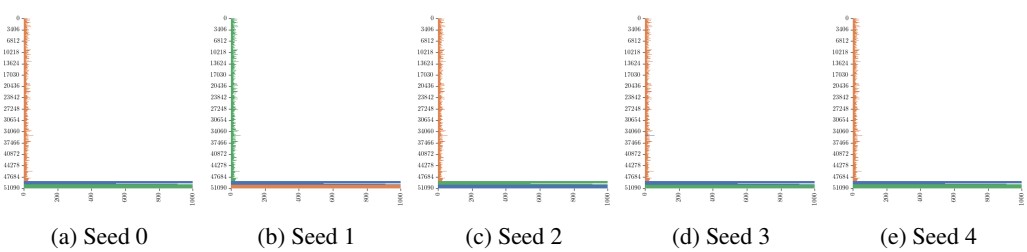

Figure 9: Clustering visualisations for walker2d-random-medium-expert-v2

## E  ADDITIONAL FIGURES

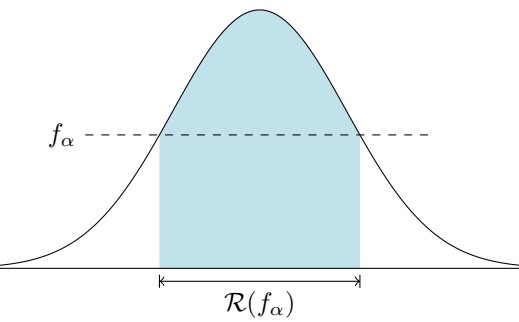

Figure 10:  A diagram showing the probability density function and the $100(1-\alpha)\,\%$ highest density region of a normal distribution. The probability of the corresponding normal random variable to lie inside $\mathcal{R}(f_\alpha)$, which corresponds to the area of the coloured region, is $1 - \alpha$.

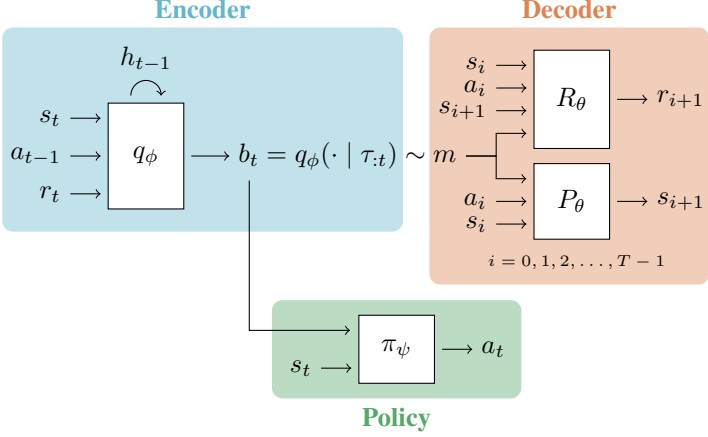

Figure 11:  VariBAD architecture. This figure is a redrawn version of Figure 2 in Zintgraf et al. (2021).

## F  RELATED WORK

**Value regularisation**    Value regularisation aims to discourage the actor from choosing OOD actions by penalising their critic values. Conservative Q learning (CQL; Kumar et al. 2020) was one of the first works in this line of research, where they minimise the standard TD error together with the Q-values of OOD actions. Lyu et al. (2022) pointed out that the CQL excessively regularsies the OOD Q-values to the extent that hampers the learning process. They suggested a milder regularisation term based on the critic values of ID actions. Supported value regularisation (SVR; Mao et al. 2023) proposed a penalisation scheme that maintains standard Bellman updates for ID actions while selectively penalising OOD actions' critic values. Most existing value regularisation algorithm, including the three works introduced in this section, sample the OOD actions from the current policy. However, as training progresses, the current policy will start to produce ID samples, so it is crucial to prevent unnecessary penalisation for those actions. CQL circumvents this issue through maximising the critic values for actions in the dataset. SVR does it by soft thresholding the regulariser based on the importance sampling ratio. In contrary, our method adopts a hard thresholding mechanism where ID actions are not penalised at all. This is possible due to our capability of explicitly identifying the OOD action set.

**Heterogeneous datasets**    There are multiple prior work concerned with offline RL datasets with heterogeneous behaviours. Wang et al. (2023) utilises a diffusion model (Sohl-Dickstein et al., 2015; Ho et al., 2020) to capture the multi-modality of the true behaviour policy. Li et al. (2023) trains a mixture of Gaussian policy on the dataset via likelihood maximisation and then obtains a closed-form estimate of the best possible action near the behaviour policy. These two works ignores the trajectory information and handles each transition individually. Mao et al. (2024) incorporates an expectation–maximisation algorithm to learn diverse policies from a given offline RL dataset. Wang et al. (2024) proposes a learning-based trajectory clustering algorithm that can also automatically determine the cluster size. Although these two works leverage the trajectory information, they obtain the trajectory representation by simply averaging the samples, causing a substantial loss of information. We incorporate a sequence modelling technique instead to learn an effective representation of each trajectory.

**VQ-VAE**    State-conditioned action quantisation (SAQ; Luo et al. 2023) is closely related to our work in the sense that they also leverage a VQ-VAE in the offline RL setting. However, their main focus is to discretise the actions because most of the challenges in offline RL originates from the ambiguity of continuous distributions. On the other hand, our algorithm uses VQ-VAE to cluster the trajectories and recover the behaviour policies. Also, SAQ discretises the actions individually, ignoring the trajectory information.

## G    LIMITATIONS

Our work is built upon the assumption that each trajectory in the dataset was sampled from a single behaviour policy. Although this assumption does not hold in general, as the behaviour policy may change mid-trajectory, the change is subtle enough for our algorithm to perform reasonably well. However, this might not be the case for real world scenarios. Future work could explore mechanisms to detect behaviour policy change and split the trajectory at those transition points.

