# OpenReview forum: "Offline Reinforcement Learning Through Trajectory Clustering and Lower Bound Penalisation"
_ICLR.cc/2026/Conference — Submitted to ICLR 2026_

### Official Review · Reviewer_kw1y · 2025-10-28

**Soundness:** 2
**Presentation:** 2
**Contribution:** 2
**Rating:** 4
**Confidence:** 2

**Summary:**

This paper proposes a value-penalized offline RL algorithm. It first identifies OOD actions (for computing the target value), i.e., whether $a' \in OOD(s')$ using a clustering method that addresses the challenges posed by heterogeneous datasets. It then applies a modified Bellman operator that replaces OOD values with lower-bound values.

The paper identifies OOD actions in heterogeneous datasets using a clustering algorithm, which appears to be directly borrowed from existing meta-RL literature, specifically variBAD [1].

The second focus of this work is the penalization of these OOD values (while the standard Bellman operator is applied to in-distribution data). The penalization is achieved by setting the target value to: $V(s) - r_{max} + r_{min} - \gamma K_V K_P D(a,a')$

The experimental section mainly focuses on MuJoCo tasks. Given that the main contribution of this paper lies in the theoretical justification, I believe it is not strictly necessary to include comparison results on other tasks.

[1] Luisa M. Zintgraf, Sebastian Schulze, Cong Lu, Leo Feng, Maximilian Igl, Kyriacos Shiarlis, Yarin Gal, Katja Hofmann, and Shimon Whiteson. VariBAD: Variational Bayes-Adaptive Deep RL via Meta-Learning. JMLR 22 (2021): 1–39.

**Strengths:**

The analysis in Section 4.2 is excellent. It proves not only the convergence of the customized Bellman operator (γ-contraction) but also the optimality of the resulting policy.

**Weaknesses:**

W1 Novelty of Section 4.1.
This section describes the method for identifying OOD actions in (more realistic) heterogeneous datasets, which seems to constitute about half of the paper’s contribution. However, this component appears to be directly borrowed from existing meta-RL literature, particularly variBAD [1]. It would be the AC’s responsibility to verify the novelty and contribution of this part to the community.

W2/Q1 Organization and clarity.
The organization of the paper is somewhat confusing. The main focus is clearly the theoretical analysis of the proposed penalized Bellman operator. However, I found it difficult to connect the theoretical formulation to its practical implementation. The related concerns are listed in the next section.

**Questions:**

How are the values of $K_V$ and $K_P$ in Equation (7) determined when computing the target value for OOD actions in the experiments?

---

> ### Author Response · Authors · 2025-11-22
>
> We thank Reviewer kw1y for their careful reading of our manuscript and for their insightful comments. We address each point in detail below.
>
> ## W1. Novelty of Section 4.1
>
> We appreciate the reviewer’s concern regarding novelty. However, our primary contribution is not the algorithm itself, but the specific problem formulation of **trajectory clustering as a Markov reward process (MRP) identification problem**. Under the assumption that the trajectories are generated by one of $K$ candidate behaviour policies indexed by a discrete latent variable $Z$, we formalise the trajectory clustering as **identifying the MRP associated with each value of $Z$**. This viewpoint naturally leads to a Bayesian inference objective in which we learn a posterior distribution of the latent variable that explains the observed trajectories.
>
> To the best of our knowledge, this **MRP-identification formulation** has not been explored in existing offline RL work. Furthermore, although our amortised inference module was inspired by the variBAD encoder, using such an encoder specifically to cluster trajectories in heterogeneous offline RL datasets, rather than to infer tasks in meta-RL, is a new application.
>
> ## W2 & Q1. Practical implementation of the algorithm
>
> In all experiments, we treat the combined hyperparameter $K=K_VK_P$ as a single tunable constant. To demonstrate robustness, we include the following sensitivity analysis over a wide range of $K$, showing that the algorithm is stable with respect to this choice.
>
> |   $K$ | halfcheetah-medium-v2   | hopper-medium-v2   | walker2d-medium-v2   |
> |:---------------------:|------------------------:|-------------------:|---------------------:|
> |                  0.1 | 62.53 ± 0.73            | 102.08 ± 2.77      | 86.88 ± 27.56        |
> |                  0.2 | 63.07 ± 1.35            | 102.6 ± 0.68       | 90.59 ± 24.99        |
> |                  0.5 | **63.48 ± 1.19**           | 102.16 ± 0.79      | 93.12 ± 26.63        |
> |                  1   | 63.43 ± 0.91            | 102.03 ± 5.72      | **96.05 ± 13.86**        |
> |                  2   | 63.13 ± 1.06            | **103.43 ± 0.89**      | 93.86 ± 15.6         |
> |                  5   | 63.33 ± 1.26            | 102.33 ± 3.91      | 92.27 ± 1.56         |
> |                 10   | 62.79 ± 0.84            | 102.46 ± 1.51      | 89.52 ± 1.54         |
> |                 20   | 63.19 ± 1.15            | 99.97 ± 6.75       | 88.04 ± 1.55         |
> |                 50   | 62.59 ± 1.01            | 84.11 ± 20.69      | 81.68 ± 9.06         |
> |                100   | 62.11 ± 1.17            | 92.47 ± 21.06      | 78.41 ± 13.7         |
>
> As shown, the performance varies smoothly across several orders of magnitude of $K$, indicating that the method is insensitive to tuning.

---

### Official Review · Reviewer_w7Vj · 2025-10-31

**Soundness:** 2
**Presentation:** 2
**Contribution:** 2
**Rating:** 4
**Confidence:** 2

**Summary:**

This paper proposes a new framework for value regularisation in offline reinforcement learning (RL) that explicitly identifies out-of-distribution (OOD) regions, rather than approximating them indirectly as prior methods do (e.g., CQL, SVR, or mild regularisation approaches).

Following are the key ideas:

1. Trajectory clustering for OOD identification: The authors develop a trajectory-level clustering algorithm that models the data as arising from multiple unknown behaviour policies. Using a meta-learning–style latent variable model inspired by VariBAD but modified with state-space layers (S5) and a VQ-VAE structure, the method assigns trajectories to clusters representing different behaviour policies. This explicit clustering allows identification of in-distribution (ID) and OOD action sets, even when the dataset is heterogeneous or multi-modal.

2. Lower Bound Penalisation (LBP): Once the OOD region is determined, the method introduces a Bellman-type operator that penalises critic Q-values of OOD actions by regressing them towards a tight lower bound of the optimal value function Q*. The lower bound is derived under Lipschitz continuity assumptions on the transition and value functions, ensuring that the penalised values remain conservative but not overly pessimistic. In-distribution actions are left unpenalised, addressing the over-regularisation issue of prior methods like CQL.

**Strengths:**

1. Novel integration of trajectory clustering into offline RL: The idea to use trajectory-level clustering to infer multiple behaviour policies is  novel, and goes beyond existing techniques such as density-based OOD estimation methods. Authors provide a new practical approach for explicitly identifying in-distribution and OOD action regions, which is novel for offline RL. While theoretical grounding may be limited, the design idea is fresh and potentially useful.

2. Empirical rigor and diversity of experimental evaluation: Algorithm is tested across both standard D4RL benchmarks and synthetically mixed datasets (combining random, medium, and expert trajectories). The reported results consistently outperform CQL, TD3+BC, and other baselines, showing strong empirical consistency.

**Weaknesses:**

Though the idea of estimating OOD regions is novel, I feel that the novelty of the paper is somewhat limited and results not surprising.

1. The paper assumes that clustering over trajectories can reliably separate distinct behaviour policies. However, there is no theorem or formal guarantee that the latent clusters correspond to true behavioural modes or that the learned clusters align with distinct data-generating policies. Without such theoretical validation, the separation between "in-distribution" and "out-of-distribution" regions is empirically plausible but not provable, which weakens the foundation.

2. Missing quantitative link between theoretical lower bound and empirical robustness: It is claimed that the proposed lower-bound penalisation leads to improved robustness and sample efficiency.
However, no analytical relation or empirical metric is provided to connect the derived lower bound to the actual generalisation behaviour or to OOD performance metrics. The validation is empirical and qualitative.

**Questions:**

1. Assumption 1 seems hard to justify. Variations in action space do not depend upon state value. However, we know that in many commonly encountered systems, the dynamics (and hence most likely differential of probabilities) do depend upon the current state.

2. Similarly, Assumption 2 the r.h.s. bound does not depend upon \beta.

3. Please point out explicitly the formal guarantee that the latent clusters correspond to true behavioural modes or that the learned clusters align with distinct data-generating policies.

4. Please point out the quantitative link between theoretical lower bound and empirical robustness.

I would be happy to increase score if 3. and 4. above are addressed.

---

> ### Author Response · Authors · 2025-11-22
>
> We thank Reviewer w7Vj for their careful reading of our manuscript and for their insightful comments. We address each point in detail below.
>
> ## Q3 & W1. Formal link between latent clusters and behavioural modes
>
> We clarify the theoretical and empirical aspects of our clustering algorithm here.
>
> ### (a) Why VAE/VQ-VAE can recover latent behavioural modes
>
> A VAE models the data distribution as infinite Gaussian mixture:
> $$p(x) = \int p(z) p(x| z) dz$$
> where $p(x|z)$ is conditional Gaussian parametrised by a high-capacity neural network. Under the variational framework, the prior p(z) is replaced with a proposal density $q(z|x)$, and $q(z | x)$ is jointly optimised with $p(x|z)$, finding relevant latent z for each x. Prior works such as Wang & Blei (2019) show that, under mild identifiability and regularity assumptions, variational inference recovers latent components that best explain the observed data.
>
> VQ-VAE further discretises $z$, effectively turning the model into a finite mixture of Gaussian policies. When behavioural trajectories come from a finite mixture of behaviour policies $\\{\beta_0, \beta_1, \dotsc, \beta_{K-1}\\}$, each policy induces a characteristic distribution over trajectories. VQ-VAE leverages this distinctiveness to align latent codes with the underlying modes.
>
> ### (b) Empirical validation
>
> Our results in Table 4 show **ARI and NMI close to 1**, which strongly indicates that latent clusters correspond to true generating policies. Both ARI and NMI are standard clustering metrics where a score of 1 represents perfect alignment with ground truth.
>
> To further strengthen this, we have added visualisations of the clustering results in Appendix D.2.
>
> **Contribution**
>
> We emphasise that VQ-VAE-based clustering of trajectories is itself a contribution of this paper. It provides a principled mechanism for identifying behaviour modes in offline RL datsets, which may be useful for downstream behaviour analysis.
>
> ## Q4 & W2. Quantitative link between lower bound and robustness
>
> We appreciate the request for a clearer connection between the theoretical bound and empirical robustness. Sample efficiency is indirectly demonstrated by strong offline performance under the same size of offline data. A more explicit demonstration would come from online fine-tuning:
>
> * If the offline Q-function is closer to the true one, the online phase requires fewer samples to converge;
> * Conversely, if OOD Q-values are overly penalised, the online phase must *undo* this distortion, reducing sample-efficiency.
>
> This mechanism links the lower-bound regularisation to improved generalisation.
> We are currently running simulations to quantify this offline-to-online improvement and will report the results as soon as they are ready.
>
> ## Q1. Justifying Assumption 1
>
> Several recent works in RL assume continuity of the transition probability kernel in the Wasserstein-1 metric (Kara & Yüksel 2023; Kara et al., 2023). Under compactness of the action space, Wasserstein-1 continuity implies Lipschitz continuity in the action variable. Moreover, Asadi et al. (2018) assume Lipschitz continuity in both state and action variables, which is stronger than our assumption. Thus, Assumption 1 is consistent with standard regularity assumptions commonly used in the RL community.
>
> ## Q2. Justifying Assumption 2
>
> We agree that the Lipschitz constant of the value function generally depends on the specific behaviour policy. For our mixture model $\\{\beta_0, \beta_1, \dotsc, \beta_{K-1}\\}$, each policy $\beta_k$ has its own Lipschitz constant $K_k$. Then we may introduce a global constant
> $$K=\max_k K_k,$$
> which upper-bounds all of the $K$ constants.
>
> ### References
>
> Asadi, Kavosh, Dipendra Misra, and Michael Littman. "Lipschitz continuity in model-based reinforcement learning." International conference on machine learning. PMLR, 2018.
>
> Kara, A. D., & Yüksel, S. (2023). Q-learning for continuous state and action MDPs under average cost criteria. arXiv preprint arXiv:2308.07591.
>
> Kara, A., Saldi, N., & Yüksel, S. (2023). Q-learning for MDPs with general spaces: Convergence and near optimality via quantization under weak continuity. Journal of Machine Learning Research, 24(199), 1-34.
>
> Wang, Y., & Blei, D. M. (2019). Frequentist consistency of variational Bayes. Journal of the American Statistical Association, 114(527), 1147-1161.

---

### Official Review · Reviewer_ZSpB · 2025-10-31

**Soundness:** 2
**Presentation:** 3
**Contribution:** 2
**Rating:** 4
**Confidence:** 4

**Summary:**

The paper proposes a framework for explicit OOD action identification in offline RL and a new lower‑bound value regularizer to penalize Q-values only outside the estimated in‑distribution (ID) set.

**Strengths:**

Most value‑regularization methods avoid explicit OOD delineation; this paper defines ID via analytically computed HDRs (Gaussian) and unions across clusters. The closed‑form HDR for Gaussians and the link to confidence regions are clear and practical.

Formulating trajectory clustering with a discrete latent "task" and training an amortized posterior (VQ‑VAE + S5 encoder) is a nice adaptation of variBAD‑style ideas to offline RL. The ablation in Table 4 and extended Table 6 shows the modeling choices (transition on, reward off) matter and supports the claim.

The derived Q-lower bound is tight and connects to Wasserstein Lipschitzness of dynamics and Lipschitz. The contraction‑mapping based results for the penalized operator are careful and readable.

Misc. Notes:
The paper offers competitive empirical results, informative figures, and the presentation + organization are sufficient.

**Weaknesses:**

The tightness and usefulness of $Q_{\beta}^{LB}$ hinge on $K_V, K_P$. I could not find how these are estimated or tuned in practice; yet they appear in the penalty target throughout (Sec. 4.2 p.5–6; Appx. C.3 p.22–23). Without principled estimation or sensitivity analysis, the regularizer may effectively collapse back to $r_{min} / 1 - \gamma$. Please describe how $K_V, K_P$ are chosen, (ii) include an ablation showing performance vs. $K_V, K_P$.

The use of extra datasets to set $r_{min}, r_{max} may break comparability. Clarify whether competing baselines also used cross‑dataset reward ranges; if not, redo with per‑dataset ranges or discuss fairness.

Table 2 appears to omit the “halfcheetah‑medium‑expert” score for the paper's method (Ours) and does not report an Average for Ours, although other methods have an average row. This impedes a balanced comparison.

Stage III samples OOD actions by drawing from a large hypercube in $tanh^{-1}$ space and rejecting ID points. In higher‑dimensional actions this can be extremely inefficient; no acceptance‑rate analysis or runtime is provided.

The theory assumes each trajectory is generated by a single stationary behavior, but many offline datasets (e.g., medium‑replay) arise from non‑stationary policies within a trajectory. The authors acknowledge this (Limitations section), yet the clustering encoder uses action‑less sequences, which may struggle when rewards are noisy and state overlaps across behaviors are large. Please (i) evaluate on truly mixed‑policy trajectories (e.g., replay‑style) with known switches, (ii) compare to encoders that also see actions.

**Questions:**

How are $K_V, K_P$ set in practice? Are they tuned per task, fixed across tasks, or estimated from data (e.g., via empirical Lipschitz upper bounds)? Please include a sensitivity figure.

Did all baselines also compute $r_{min}, r_{max}$ across the union of datasets, or only from the target dataset? If not, could you rerun with per‑dataset ranges?

---

> ### Author Response · Authors · 2025-11-22
>
> We thank Reviewer ZSpB for their careful reading of our manuscript and for their insightful comments. We address each point in detail below.
>
> ## W1 & Q1. Choice of $K_V, K_P$ and their sensitivity
>
> We set $K=K_VK_P$ as a single tunable hyperparameter. We have added a sensitivity analysis on three different datasets: halfcheetah-medium-v2, hopper-medium-v2, walker2d-medium-v2. Performance remains stable across a wide range of $K$, indicating that the penalty does not collapse to $r_\min / (1-\gamma)$ in practice.
>
> |   $K$ | halfcheetah-medium-v2   | hopper-medium-v2   | walker2d-medium-v2   |
> |:---------------------:|------------------------:|-------------------:|---------------------:|
> |                  0.1 | 62.53 ± 0.73            | 102.08 ± 2.77      | 86.88 ± 27.56        |
> |                  0.2 | 63.07 ± 1.35            | 102.6 ± 0.68       | 90.59 ± 24.99        |
> |                  0.5 | **63.48 ± 1.19**           | 102.16 ± 0.79      | 93.12 ± 26.63        |
> |                  1   | 63.43 ± 0.91            | 102.03 ± 5.72      | **96.05 ± 13.86**        |
> |                  2   | 63.13 ± 1.06            | **103.43 ± 0.89**      | 93.86 ± 15.6         |
> |                  5   | 63.33 ± 1.26            | 102.33 ± 3.91      | 92.27 ± 1.56         |
> |                 10   | 62.79 ± 0.84            | 102.46 ± 1.51      | 89.52 ± 1.54         |
> |                 20   | 63.19 ± 1.15            | 99.97 ± 6.75       | 88.04 ± 1.55         |
> |                 50   | 62.59 ± 1.01            | 84.11 ± 20.69      | 81.68 ± 9.06         |
> |                100   | 62.11 ± 1.17            | 92.47 ± 21.06      | 78.41 ± 13.7         |
>
> ## W2 & Q2. Cross-dataset reward ranges
>
> SVR also uses cross-dataset ranges. For completeness, we reran our method with per-dataset ranges and include the results below. The difference of $\Delta r=r_\max - r_\min$ between the two settings is less than 5% of the average value scale, and the performance differences are correspondingly small. We used cross-dataset ranges mainly to share hyperparameters between medium-expert and expert-datasets, which improves reproducibility.
>
> | cross_dataset   | halfcheetah-medium-v2   | hopper-medium-v2   | walker2d-medium-v2   |
> |:---------------------------:|------------------------:|-------------------:|---------------------:|
> | True                      | **63.48 ± 1.19**            | **103.43 ± 0.89**      | 96.05 ± 13.86        |
> | False                       | 63.02 ± 0.68            | *101.54 ± 1.57*      | **96.53 ± 18.08**        |
>
> On hopper-medium-v2, per dataset ranges produced relatively large $r_\min/(1-\gamma)$, causing weak regularisation and critic divergence. We disabled reward normalisation for this dataset to avoid instability.
>
> ## W3. Missing value in Table 2
>
> We appreciate the reviewer for noting this inconsistency. The missing entries have been fixed in the updated pdf.
>
> ## W4. OOD sampling efficiency
>
> The reviewer is correct that uniform-hypercube sampling becomes inefficient in high-dimensional action spaces. However, we emphaisise that **none of the experiments reported in the submission use hypercube sampling**. All results in the PDF, including those in the main table, were produced using the following **mixture distribution**:
> $$0.5 \mathcal{N}(\mu(s), \kappa^2\Sigma(s)) + 0.5 \pi(\cdot \mid s),$$
> where $\mu$ and $\Sigma$ approximate the behaviour policy and $\kappa=2$. This sampling strategy yields substantially better empirical performance, particularly on the expert datasets.
>
> The reference to uniform hypercube sampling in Appendix C.3 was an **outdated artifact** from an early draft. We unintentionally missed updating that paragraph prior to submission. The revised version now correctly describes the mixture-based method that was used for the experiments. We thank the reviewer for identifying this.

---

> ### Author Response · Authors · 2025-11-22
>
> ## W5. Mixed-policy trajectories and encoders with actions
>
> To evaluate on a truly mixed-policy dataset with known switches, we constructed a custom [HalfCheetah-v5](https://gymnasium.farama.org/environments/mujoco/half_cheetah/) dataset using two distinct behavior policies provided by the Minari repository:
>
> * **Expert policy**: [TQC-expert](https://huggingface.co/farama-minari/HalfCheetah-v5-TQC-expert)
> * **Medium policy**: [TQC-medium](https://huggingface.co/farama-minari/HalfCheetah-v5-TQC-medium)
>
> We generated 1000 trajectories, divided into four types:
>
> 1. **Expert-only trajectories:**
>
>     1000-step trajectories generated entirely by the expert policy.
>
> 2. **Medium-only trajectories:**
>
>     1000-step trajectories generated entirely by the medium policy.
>
> 3. **Expert → Medium mixed trajectories:**
>
>     First 500 steps generated by the expert policy, followed by 500 steps generated by the medium policy.
>
> 4. **Medium → Expert mixed trajectories:**
>
>     First 500 steps generated by the medium policy, followed by 500 steps generated by the expert policy.
>
> Thus, **half of the dataset contains mid-trajectory policy switches**, violating the single-trajectory single-policy assumption.
>
> Before applying our clustering algorithm, we performed a simple preprocessing step:
> **each 1000-step trajectory was segmented into two 500-step sub-trajectories.**
> This ensures that each sub-trajectory corresponds to a stationary behaviour policy. We emphasise that:
>
> * this segmentation is trivial (fixed-length splitting, no oracle labels),
>
> * and it does not require knowledge of the underlying policies.
>
> After segmentation, the dataset consists of 2000 sub-trajectories, each drawn from either the expert or medium policy.
>
> Our trajectory clustering module achieves:
>
> * ARI: 0.95 ± 0.04
>
> * NMI: 0.91 ± 0.06
>
> and the final policy learned by our framework achieves a normalised return of **17386 ± 116**, outperforming the average return of the expert-only trajectories (**16227 ± 955**).
>
> We also evaluated alternative encoders that include actions as input. Across six custom datasets, **action-aware encoders perform worse on average** (ARI 0.97 vs. 0.99; NMI 0.97 vs 0.98; see Table 7 on page 26), supporting our design choice of using state-only encoders.
>
> | dataset                                    | ARI         | NMI         |
> |:-------------------------------------------|------------:|------------:|
> | halfcheetah-custom-medium-expert-v2        | 1.0 ± 0.0   | 0.99 ± 0.0  |
> | halfcheetah-custom-random-medium-expert-v2 | 0.99 ± 0.0  | 0.99 ± 0.01 |
> | hopper-custom-medium-expert-v2             | 1.0 ± 0.0   | 1.0 ± 0.0   |
> | hopper-custom-random-medium-expert-v2      | 0.82 ± 0.24 | 0.88 ± 0.15 |
> | walker2d-custom-medium-expert-v2           | 1.0 ± 0.0   | 1.0 ± 0.0   |
> | walker2d-custom-random-medium-expert-v2    | 0.98 ± 0.04 | 0.98 ± 0.04 |
> | *Average* | 0.97 ± 0.11 | 0.97 ± 0.07 |

---

> > ### Comment · Reviewer_ZSpB · 2025-11-24
> > **Thank you**
> >
> > Thank you for clarifying the points I made. I believe these clarifications and experiments warrant me to raise my score.

---

> > > ### Author Response · Authors · 2025-11-24
> > >
> > > Dear Reviewer ZSpB
> > >
> > > Thank you very much for your re-evaluation and raising the score.
> > > We appreciate your time and efforts.
> > >
> > > Authors

---

### Official Review · Reviewer_3gtC · 2025-10-31

**Soundness:** 3
**Presentation:** 2
**Contribution:** 2
**Rating:** 4
**Confidence:** 3

**Summary:**

The authors point out that many existing offline RL methods sidestep explicit OOD detection and instead rely on indirect conservatism (e.g., value or policy penalties), which can break down when the logged data come from a multi-modal behavior distribution. To address this, the paper introduces a trajectory-based behavior modeling approach that clusters trajectories and uses the resulting per-cluster behavior policies to explicitly characterize the in-distribution and OOD region. On top of this, the authors derive a Bellman-style operator that replaces or pushes down Q-values for OOD actions using a theoretically motivated lower bound, thereby reducing overestimation in unsupported regions. Empirically, they evaluate on standard D4RL benchmarks and on a constructed heterogeneous dataset, reporting improvements over representative offline RL baselines.

**Strengths:**

1. Many offline RL methods do not explicitly determine which state–action pairs are OOD, but instead apply global/indirect conservatism, which can inadvertently penalize valid but low-frequency actions when the data happen to be heterogeneous or multi-modal.

2. This paper proposes a method to prevent overestimation of the Q-values for OOD state–action pairs in offline RL. Instead of directly penalizing them, the authors first identify the OOD regions and then introduce a new Bellman operator to push down the values of the state–action pairs in those regions.

3. The paper is well structured and easy to follow

**Weaknesses:**

1. Problem setup not fully validated. The paper’s main motivation is that existing OOD-penalization methods can fail on heterogeneous / multi-modal datasets. However, the authors also note in Section 5.1 that many of the standard offline-control benchmarks they use (including several D4RL tasks) are effectively closer to uni-modal at the state–action level. Since the paper does not quantify how heterogeneous the actual benchmarks are, it is hard to tell how often the proposed clustering-based OOD identification is really needed.

2. The positive-definite covariance assumption is strong. Around line 163 the paper assumes that, for every state  $s$, the behavior policy’s covariance $\Sigma(s)$ is positive definite. For offline data, especially when some states are rarely visited or when actions concentrate on a low-dimensional manifold, this is not guaranteed. The paper should at least discuss how to handle rank-deficient / ill-conditioned covariances.

3. Theory relies on strong global continuity assumptions. The main Bellman-type result depends on 1-Wasserstein continuity of the dynamics in the action space and Lipschitz continuity of the value in the state space. While these assumptions are intuitively reasonable (“nearby actions/states behave similarly”), they are fairly strong and may not hold in environments with mode switches or highly non-smooth rewards. The paper should provide evidence or discussion about how broadly these assumptions apply in common offline RL benchmarks.

4. Missing key ablations. The method critically depends on correctly identifying the number of behavior modes $K$ and on the quality of the trajectory-based behavior model, but there is no ablation on mis-specified  $K$, on different HDR levels, or on simpler clustering baselines. This makes it difficult to assess robustness.

5. Complexity vs. gains is unclear. Compared with prior offline RL methods, the proposed approach introduces extra components (sequence/trajectory modeling, VQ-VAE, per-cluster behavior estimation) to enable explicit OOD detection, but the reported performance improvements over strong baselines are modest. A clearer comparison of computational/training complexity versus returned performance would strengthen the practicality claim.

6. Limited evidence for real/practical settings. Because OOD identification here is tied to the learned VAE-style behavior model and is only evaluated on MuJoCo-style (and partly synthetic/heterogeneous) datasets, it is not clear how well the approach would scale to larger or noisier real-world logs, where behavior is messier and coverage is sparser. A discussion or experiment in a more realistic domain would be helpful.

**Questions:**

Please refer to the weakness part.

---

> ### Author Response · Authors · 2025-11-22
>
> We thank Reviewer 3gtC for their careful reading of our manuscript and for their insightful comments. We address each point in detail below.
>
> ## W1 & W6. On Dataset Heterogeneity and the Limitations of the Current Setup
>
> We acknowledge that the current benchmarks do not fully validate the motivation of our method. Although heterogeneous and multi-model datasets motivated this work, our experiments rely on MuJoCo datasets that are effectively uni-modal at the state-action level. As stated in the Limitations section, the proposed approach assumes a single-behaviour for each trajectory. Our recent (unsubmitted) investigations indicate that this assumption is often violated in more realistic datasets, where the behaviour policy changes mid-trajectory. In such cases, our method fails to yield a competent policy. However, when the change points are known, clustering sub-trajectories instead of the whole trajectories resolves this issue.
>
> To examine this concretely, we constructed a custom [HalfCheetah-v5](https://gymnasium.farama.org/environments/mujoco/half_cheetah/) dataset using two distinct behaviour policies from the Minari repository ([medium](https://huggingface.co/farama-minari/HalfCheetah-v5-TQC-medium) and [expert](https://huggingface.co/farama-minari/HalfCheetah-v5-TQC-expert)). In half of the trajectories, we manually introduced a mid-trajectory policy switch. After applying a simple fixed-length segmentation into sub-trajectories, our clustering module achieves ARI 0.95 ± 0.04 and NMI 0.91 ± 0.06, and the resulting policy obtains a return of **17386 ± 116**, exceeding the expert-only trajectory average (**16227 ± 955**).
>
> In practical applications, change points are not known in advance. We are therefore exploring **unsupervised change-point detection (CPD)** as a natural extension of our method. This line of work is in its early stages, and we do not yet have any results to report, but CPD offers a principled way to relax the single-policy assumption. We are also extending our evaluation to more complex datasets such as D4RL antmaze and OGBench(Park et al., 2024).
>
> *(See Section W5 of our [response to Reviewer ZSpB](https://openreview.net/forum?id=DOL96nnJpm&noteId=0cXyozJDH7) for dataset-construction details.)*
>
> ## W2. Positive-definite covariance assumption
>
> We agree that real-world behaviour data may exhibit low-rank or concentrated action distributions. Our assumption of positive-definite behaviour covariances follows standard practice in offline RL, where Gaussian policies with parametrised diagonal covariances are commonly adopted (e.g., CQL, IQL, MCQ, SVR). Such parametrisations enforce strictly positive variance through softplus/exponential mappings.
>
> That said, our framework is compatible with dimension reduction techniques such as principal component analysis (PCA), which can be applied before covariance estimation. In cases where empirical covariances are rank-deficient, our method can be applied in a reduced action subspace using PCA.
>
> ## W3. Continuity assumptions
>
> Several recent works in RL assume continuity of the transition probability kernel in the Wasserstein-1 metric (Kara & Yüksel 2023; Kara et al., 2023). Under compact action spaces, this implies Lipschitz continuity in the action variable. Moreover, Asadi et al. (2018) assume Lipschitz continuity in both state and action variables, which is stronger than our assumption. The Lipschitz continuity of the value function is also a widely accepted assumption (Tang et al., 2020; Zhang, 2022).
>
> These continuity conditions formalise the intuitive requirement that small perturbations in actions or states result in similar next-state distributions and rewards, a property satisfied by standard benchmarks with smooth dynamics and reward functions. Importantly, our method does not require assumptions beyond those commonly used in theoretical analyses of offline RL. In environments with abrupt mode switches or non-smooth rewards, any value-function approximator would face similar challenges.

---

> ### Author Response · Authors · 2025-11-22
>
> ## W4. Missing ablations
>
> Below we include ablations on the OOD threshold parameter α. Performance is stable across a wide range of α, demonstrating that the method is not sensitive to the choice. Note that we used α to 0.5 for all of our experiments in the submission.
>
> |  α  | halfcheetah-medium-v2   | hopper-medium-v2   | walker2d-medium-v2   |
> |:------------------:|------------------------:|-------------------:|---------------------:|
> |               0.1 | 63.2 ± 1.12             | 102.15 ± 0.51      | 92.97 ± 22.93        |
> |               0.3 | 62.38 ± 1.35            | 103.25 ± 0.86      | 91.33 ± 22.36        |
> |               0.5 | **63.48 ± 1.19**            | **103.43 ± 0.89**      | 96.05 ± 13.86        |
> |               0.7 | 63.34 ± 0.98            | 102.36 ± 0.71      | 94.16 ± 20.35        |
> |               0.9 | 62.45 ± 1.22            | 103.13 ± 0.51      | **96.96 ± 15.2**         |
>
> Regarding K, our experiments already use a misspecified K=4 across all datasets, and the method's performance was empirically robust to this choice.
>
> ## W5. Concerns on the computational complexity of the algorithm
>
> Our method adds a lightweight VQ-VAE+S5 module whose training requires less than 10 minutes on a single RTX 4090 GPU. Importantly, at evaluation time our policy requires only a few MLP forward passes. This leads to significantly lower inference complexity compared to diffusion- or flow-based offline RL methods, which rely on iterative sampling processes.
>
> ## References
>
> Asadi, Kavosh, Dipendra Misra, and Michael Littman. "Lipschitz continuity in model-based reinforcement learning." International conference on machine learning. PMLR, 2018.
>
> Kara, A. D., & Yüksel, S. (2023). Q-learning for continuous state and action MDPs under average cost criteria. arXiv preprint arXiv:2308.07591.
>
> Kara, A., Saldi, N., & Yüksel, S. (2023). Q-learning for MDPs with general spaces: Convergence and near optimality via quantization under weak continuity. Journal of Machine Learning Research, 24(199), 1-34.
>
> Park, S., Frans, K., Eysenbach, B., & Levine, S. (2024). Ogbench: Benchmarking offline goal-conditioned rl. arXiv preprint arXiv:2410.20092.
>
> Tang, Z., Feng, Y., Zhang, N., Peng, J., & Liu, Q. (2020). Off-policy interval estimation with lipschitz value iteration. Advances in Neural Information Processing Systems, 33, 7887-7897.
>
> Zhang, S. (2022). Conservative dual policy optimization for efficient model-based reinforcement learning. Advances in neural information processing systems, 35, 25450-25463.

---

### Official Review · Reviewer_bjZM · 2025-11-01

**Soundness:** 3
**Presentation:** 2
**Contribution:** 3
**Rating:** 6
**Confidence:** 2

**Summary:**

This paper proposes a novel framework for value regularization in offline Reinforcement Learning (RL) that directly addresses the challenge of Out-of-Distribution (OOD) value overestimation by explicitly identifying the OOD region and imposing a tighter lower bound penalty.

**Strengths:**

1. While previous methods avoid identifying the OOD region directly, this work introduces a methodology to explicitly define it, acknowledging that this region can be non-convex. The core mechanism uses the Highest Density Region (HDR) concept, a rigorous statistical tool, to set an adaptive likelihood threshold for OOD status, which is a highly original approach in RL.

2. The use of a specialized trajectory clustering algorithm, which reformulates the clustering problem as a task inference problem akin to meta RL (variBAD), is highly innovative. This approach is tailored to heterogeneous datasets and utilizes sequence modeling (S5 architecture) to learn effective trajectory representations, avoiding the information loss associated with simple averaging used in prior work.

3. The paper provides formal theoretical results, including proving the existence of a unique fixed point for the proposed penalized Bellman optimality operator.

**Weaknesses:**

1. The entire framework is built upon a fundamental assumption that may not hold true in real-world scenarios. The work assumes that each trajectory in the dataset was sampled from a single, uni-modal behavior policy.

2. The adaptive two-phase training paradigm used to automatically determine the number of clusters is a practical heuristic. It relies on a manually set threshold for removing clusters, adding another layer of tuning that is not theoretically derived.

**Questions:**

1. The selection of the confidence level $\alpha$ for the HDR is crucial yet left unspecified beyond a general principle. How sensitive is the final policy performance (e.g., average normalized score) to the choice of $\alpha$? Can the authors provide an ablation study on $\alpha$ across different D4RL domains? This is vital for showing the method's practical robustness.

---

> ### Author Response · Authors · 2025-11-22
>
> We thank Reviewer bjZM for their careful reading of our manuscript and for their insightful comments. We address each point in detail below.
>
> ## W1. On the single-trajectory single-policy assumption
>
> We agree that trajectories in real-world datasets may not always be generated by a single stationary behaviour policy. However, we view it as a practical modelling abstraction that allows theoretically-grounded behaviour clustering. In fact, our method remains robust even when this assumption is only approximately satisfied.
>
> To examine this concretely, we constructed a custom [HalfCheetah-v5](https://gymnasium.farama.org/environments/mujoco/half_cheetah/) dataset using two distinct behaviour policies from the Minari repository ([medium](https://huggingface.co/farama-minari/HalfCheetah-v5-TQC-medium) and [expert](https://huggingface.co/farama-minari/HalfCheetah-v5-TQC-expert)). In half of the trajectories, we manually introduced a mid-trajectory policy switch. After applying a simple fixed-length segmentation into sub-trajectories, our clustering module achieves ARI 0.95 ± 0.04 and NMI 0.91 ± 0.06, and the resulting policy obtains a return of **17386 ± 116**, exceeding the expert-only trajectory average (**16227 ± 955**).
>
> While such switch points are unknown In real-world datasets, this example motivates a natural extension: incorporating **unsupervised change-point detection (CPD)** to automatically identify policy shifts within trajectories. We are currently integrating CPD into our algorithm and expanding evaluation to more complex datasets such as D4RL antmaze and OGBench(Park et al., 2024). This direction provides a principled way to further relax the single-trajectory single-policy assumption without altering our core framework.
>
> Finally, it is worth noting that in many practical domains, such as user interaction logs, each trajectory often reflects a coherent mode of behaviour, making the trajectory-level assumption a reasonable and practically useful abstraction. In such settings, our method provides what we believe is the first **theoretically grounded** solution for trajectory clustering in offline RL.
>
> *(See Section W5 of our [response to Reviewer ZSpB](https://openreview.net/forum?id=DOL96nnJpm&noteId=0cXyozJDH7) for dataset-construction details.)*
>
> ## W2. On the adaptive two-phase training and thresholding
>
> The purpose of the threshold is to prevent clusters with insufficient trajectory support from degrading the learned behavior models. Although the two-phase paradigm is heuristic, in practice it is substantially less sensitive than fixing the number of clusters a priori, which typically requires careful tuning and prior knowledge.
>
> We used a **threshold of 0.1 across all experiments** without any hyperparameter search. Empirically, we observed that clusters removed during the first phase correspond to those that the clustering network naturally learns to ignore. Thus the threshold primarily formalizes an effect that already occurs implicitly.
>
> ## Q1. On the sensitivity to the HDR confidence level α
>
> We used α = 0.5 in all of our experiments. To study its sensitivity, we provide a ablation over five different values of α (0.1, 0.3, 0.5, 0.7, 0.9) on three D4RL datasets:
>
> |  α  | halfcheetah-medium-v2   | hopper-medium-v2   | walker2d-medium-v2   |
> |:------------------:|------------------------:|-------------------:|---------------------:|
> |               0.1 | 63.2 ± 1.12             | 102.15 ± 0.51      | 92.97 ± 22.93        |
> |               0.3 | 62.38 ± 1.35            | 103.25 ± 0.86      | 91.33 ± 22.36        |
> |               0.5 | **63.48 ± 1.19**            | **103.43 ± 0.89**      | 96.05 ± 13.86        |
> |               0.7 | 63.34 ± 0.98            | 102.36 ± 0.71      | 94.16 ± 20.35        |
> |               0.9 | 62.45 ± 1.22            | 103.13 ± 0.51      | **96.96 ± 15.2**         |
>
> Across all datasets, the average normalised score varies within a narrow range, demonstrating that our method is **not sensitive** to α.
>
> ## References
>
> Park, S., Frans, K., Eysenbach, B., & Levine, S. (2024). Ogbench: Benchmarking offline goal-conditioned rl. arXiv preprint arXiv:2410.20092.

---

### Meta-Review · Area_Chair_5ngv · 2026-01-05

**Summary:**

Reviewers mostly concern about (1) assumptions of the paper; (2) experiment evaluations; (3) novelty compared to meta learning. I believe that (1) and (3) are largely addressed. However, the experiment evaluations are not fully addressed, especially it lacks analysis on multi-model dataset, which should be the central contribution of the paper. Hence I recommend rejection.

**Reviewer Concerns:**

Reviewer bjZM concerns about the data collection assumption, two-phase algorithm, and the confidence level.
Reviewer 3gtC and ZSpB questions on the technical assumptions and some details of the experiments.
Reviewer w7Vj questions on an assumption and the connection between the theoretical lower bound and empirical robustness.
Reviewer kw1y mainly questions about the novelty compared to meta RL. After reading authors' rebuttal, I agree with the authors that this work has its own contribution.


However, the following concerns are not fully addressed:

1. The experiment evaluations under multi-model dataset, which is the central motivation of the work. As most reviewer pointed out, this evaluation should be an important part. During the rebuttal, authors added an experiment, which, however, is not sufficient to validate the major claims.

2. The computational cost: authors claim that the algorithm merely adds a lightweight module, which is less straightforward and evidenced. Computation time should also be compared and reported.

3. The connection between theoretical lower bound and empirical robustness are not empirically verified, and the discussion in rebuttal is rather high-level.

**Reviewer Scores:**

Please refer to the previous part.

---

### Decision · Program_Chairs · 2026-01-26

Reject